

**A novel formation mechanism of NH$_2$SO$_3$H and its enhancing effect**
**on methanesulfonic acid-methylamine aerosol particle formation in**
**agriculture-developed and coastal industrial areas**
**Hui Wang [a,‡], Shuqin Wei [a,‡], Jihuan Yang [a], Yanlong Yang [a], Rongrong Li [a], Rui Wang [a],**
**Chongqin Zhu [b,*], Tianlei Zhang [a,*], Changming Zhang [c]**
[a] *Shaanxi Key Laboratory of Catalysis, School of Chemical & Environment Science, Shaanxi*
*University of Technology, Hanzhong, Shaanxi 723001, P. R. China*
[b] *College of Chemistry, Key Laboratory of Theoretical & Computational Photochemistry of*
*Ministry of Education, Beijing Normal University, Beijing 100190, China.*
[c] *School of Mechanical Engineering, Shaanxi University of Technology, Hanzhong, Shaanxi 723001,*
*P. R. China*
## Abstract
Sulfamic acid (SFA) significantly impacts atmospheric pollution and poses potential risks to human
health. Although traditional source of SFA and its role on sulfuric acid-dimethylamine new particle
formation (NPF) has received increasing attention, the formation mechanism of SFA from HNSO$_2$
hydrolysis with CH$_3$SO$_3$H and its enhancing effect on methanesulfonic acid-methylamine APF has
not been studied, which will limit the understanding for the source and loss of SFA in agriculture-
developed and coastal industrial areas. Here, the gaseous and interfacial formation of SFA from
HNSO$_2$ hydrolysis with CH$_3$SO$_3$H was investigated using quantum chemical calculations and
BOMD simulations. Furthermore, the role of SFA in CH$_3$SO$_3$H-CH$_3$NH$_2$ system was assessed using
the Atmospheric Cluster Dynamics Code kinetic model. Our simulation results indicate that the
gaseous SFA formation from the hydrolysis of HNSO$_2$ with CH$_3$SO$_3$H can be competitive with that
catalyzed by H$_2$O within an altitude of 5-15 km. At the air-water interface, two types of reactions,
the ions forming mechanism and the proton exchange mechanism to form NH$_2$SO$_3^-$···H$_3$O$^+$ ion pair
were observed on the timescale of picosecond. Considering the overall environment of sulfuric acid
emission reduction, the present findings suggest that SFA may play a significant role in NPF and
the growth of aerosol particle as *i*) SFA can directly participate in the formation of CH$_3$SO$_3$H-
CH$_3$NH$_2$-based cluster and enhance the rate of NPF from these clusters by approximately 10$^3$ times
at 278.15 K; and *ii*) the NH$_2$SO$_3^-$ species at the air-water interface can attract gaseous molecules to
the aqueous surface, and thus promote particle growth.

* Corresponding authors, Tel: +86-0916-2641083, Fax: +86-0916-2641083.
E-mail: cqzhu@bnu.edu.cn (C. Q. Zhu), ztianlei88@l63.com (T. L. Zhang).
‡ These authors contributed equally to this work.



## 1 Introduction

As a well-studied nitrogen derivative of sulfuric acid (Rennebaum et al., 2024), sulfamic acid ($NH_2SO_3H$) was not only recognized as a potent aerosol and cloud nucleating agent (Xue et al., 2024; Zhang et al., 2023; Pszona et al., 2015; Li et al., 2018), but also can harm human health through atmospheric deposition into water bodies (Van Stempvoort et al., 2019). In agriculture-developed and industrial areas with high ammonia ($NH_3$) concentrations, such as the Yangtze River Delta in China (Yu et al., 2020), Indo-Gangetic Plains (Kuttippurath et al., 2020), Pakistan, Bangladesh (Warner et al., 2016), and the southern Italy (Tang et al., 2021), the atmospheric concentration of $NH_2SO_3H$ was expected to reach up to$10^8$ molecules·$cm^{-3}$ (Li et al., 2018), and thus lead to it becoming a significant air pollutant. So, the sources of $NH_2SO_3H$ in the atmosphere have been well investigated (Lovejoy and Hanson, 1996; Pszona et al., 2015; Li et al., 2018; Larson and Tao, 2001; Manonmani et al., 2020; Zhang et al., 2022). The traditional source of $NH_2SO_3H$ was mainly taken from the ammonolysis of $SO_3$ (Lovejoy and Hanson, 1996; Larson and Tao, 2001; Li et al., 2018). Experimentally, the rate coefficient for the ammonolysis of $SO_3$ was detected to be $2.0 \times 10^{-11}$ $cm^3$·$molecules^{-1}$·$s^{-1}$ at 295 K (Lovejoy and Hanson, 1996), which was close to the value for the hydrolysis of $SO_3$ assisted by water molecule ($10^{-11}$-$10^{-10}$ $cm^3$ $molecule^{-1}$ $s^{-1}$) (Kim et al., 1998; Hirota et al., 1996; Shi et al., 1994). Theoretically, the ammonolysis of $SO_3$ to produce $NH_2SO_3H$ can be catalyzed by $NH_3$. In arid and heavily polluted regions with high $NH_3$ concentrations, the effective rate coefficient for the ammonolysis of $SO_3$ can be sufficiently rapid, making it competitive with the conventional loss pathway of $SO_3$ with water (Li et al., 2018).

In addition to the ammonolysis of $SO_3$, new sources of $NH_2SO_3H$ formation have received increasing attention (Zhang et al., 2022; Manonmani et al., 2020, Li et al., 2018, Xue et al., 2024). The existence of $HNSO_2$ was proposed in the reaction between $SO_3$ and $NH_3$, and was regarded as the most stable for nine different isomers of $HNSO_2$, HONSO, HOSNO, HOS(O)N, $HSNO_2$, HSONO, HON(O)S, HOOSN, and HOONS (Deng et al., 2016). Owing to its similarity with $SO_3$ and the potential role of $SO_3$ in the atmosphere, the hydrolysis of $HNSO_2$ to produce $NH_2SO_3H$ formation has been focused by several groups (Zhang et al., 2022; Manonmani et al., 2020). As the direct hydrolysis of $HNSO_2$ with a high energy barrier takes place hardly in the gas phase (Zhang et al., 2022; Manonmani et al., 2020), the addition of a second water molecule (Manonmani et al.,





2020), formic acid and sulfuric acid ($H_2SO_4$, SA) (Zhang et al., 2022) have been proved to promote
the product of $NH_2SO_3H$ through the hydrolysis of $HNSO_2$. However, to the best of our knowledge,
the gaseous hydrolysis of $HNSO_2$ with $CH_3SO_3H$ has not yet been investigated. It was noted that,
with the global reduction in the concentration of $H_2SO_4$ resulting from $SO_2$ emission restrictions,
the contribution of $CH_3SO_3H$ to aerosol nucleation has received the widespread attention of
scientists. As a major inorganic acidic air pollutant (Chen et al., 2020), the concentration of
$CH_3SO_3H$ in the atmosphere was noted to be notably high across various regions, spanning from
coastal to continental, with levels found to be between 10% and 250% of those measured for SA
(Shen et al., 2019; Dawson et al., 2012; Bork et al., 2014; Shen et al., 2020; Berresheim et al., 2002;
Hu et al., 2023). Thus, understanding the hydrolysis of $HNSO_2$ with $CH_3SO_3H$ in the gas phase was
necessary for exploring its impact on aerosols and human health.
$$HNSO_2 + H_2O + CH_3SO_3H \rightarrow NH_2SO_3H + CH_3SO_3H \tag{1}$$
As a supplement to gas-phase reactions, interfacial reactions at the air-water interface not only
can accelerate the rates of atmospheric reactions but also may introduce new mechanisms (Freeling
et al., 2020; Zhong et al., 2019). For instance, the Criegee intermediates reacting with $CH_3SO_3H$ at
the air-water interface can form the ion pair of $CH_3C(H)(OOH)(SO_3CH_3)$ anhydride and $H_3O^+$ (Ma
et al., 2020), which differs from the corresponding gaseous reaction where the $CH_3SO_3H$ molecule
acts solely as a reactant reacting with Criegee intermediates directly. As far as we know, $HNSO_2$
exhibit a significant interfacial preference, as the fact that the total duration time of $HNSO_2$ at the
interface approximately accounts for 49.1% of the 150 ns simulation time (Fig. S1). However, the
hydrolysis of $HNSO_2$ with $CH_3SO_3H$ has not been studied at the air-water interface, which will
confine the understanding for the source of $NH_2SO_3H$ in regions with significant pollution and high
levels of $CH_3SO_3H$.
From a structural point of view, two functional groups of $-NH_2$ and $-SO_3H$ in the $NH_2SO_3H$
molecule can act as both hydrogen donors and acceptors to interact with atmospheric species.
Previous studies have demonstrated that SFA has a potential role in new particle formation (NPF),
as it not only clusters efficiently with itself and SA (Lovejoy and Hanson, 1996), but also can
promote the nucleation rate of NPF initiated from SA-DMA by a factor of two in dry and severely
contaminated areas with $NH_3$ (Li et al., 2018). Due to the concentration of SA in the atmosphere has
decreased significantly with the scenario of $SO_2$ emission control measures, MSA-driven NPF has



attracted growing attention (Dawson et al., 2012; Nishino et al., 2014; Chen and Finlayson-Pitts,
2017; Chen et al., 2020; Shen et al., 2020). Initially, the binary nucleation of MSA with inorganic
ammonia and organic amines in the atmosphere has been reported, where MA exhibits the strongest
enhancing capability (Chen et al., 2016; Chen and Finlayson-Pitts, 2017; Shen et al., 2019; Hu et
al., 2023). Subsequently, some reported results suggested that the triadic MSA-MA-driven NPF can
exhibit greater nucleation rates competed to the binary of MSA-driven (Zhang et al., 2022; Hu et
al., 2023). For example, both formic acid (Zhang et al., 2022) and trifluoroacetic acid (Hu et al.,
2023) exhibit an excellent catalytic influence on MSA-MA-driven NPF. However, the SFA
involved in MSA-MA-driven NPF has not been investigated, which is worth important to
investigate whether SFA can exhibit a similar enhancing effect in MSA-MA as observed in SA-
DMA.

Herein, this work studied the catalytic effect of SFA on $HNSO_2$ hydrolysis and MSA-MA

nucleation particle formation. Specifically, quantum chemical calculations were used firstly to
assess the atmospheric processes of the gaseous hydrolysis of $HNSO_2$ with $CH_3SO_3H$. Then, the
gaseous and interfacial mechanisms differences of the $HNSO_2$ hydrolysis with MSA were
investigated applying the Born-Oppenheimer Molecular Dynamic (BOMD) simulation method.
Finally, the atmospheric implications and mechanism of SFA in the MSA-MA-dominated NPF
process have been evaluated through density functional theory and the Atmospheric Clusters
Dynamic Code (ACDC) models to evaluate the potential effect of SFA on nucleation and NPF. This
work will not only deepen our understanding of the source of SFA, but also reveal significant
implications for new particle formation and aerosol particle growth in MSA polluted areas.

## 2   Methodology

### 2.1 Quantum Chemical Calculations

The gaseous hydrolysis of $HNSO_2$ with $CH_3SO_3H$ was comprehensively studied through

quantum chemistry simulations. Optimization of all the species were carried out by using the method
of M06-2X with 6-311++G(2*df*,2*pd*) basis set (Zhao and Truhlar, 2008; Elm et al., 2012; Bork et al.,
2014). Vibrational frequencies were subsequently computed at the M06-2X/6-311++G(2*df*,2*pd*)
level to ensure the reality of all stationary point's frequencies and the presence of only one imaginary
frequency in transition states. Also, at the same level, internal reaction coordinate (IRC) analyses



were conducted to verify the connection from the transition states to the corresponding products (or
reactants). All calculations regarding for geometries and frequency were conducted with the aid of
the Gaussian 09 (Frisch, 2009) program. Furthermore, to enhance the precision of the computed
energy values, single point energies were performed at the CCSD(T)-F12/cc-pVDZ-F12 (Kendall
et al., 1992; Adler et al., 2007) level utilizing the ORCA (Neese, 2012) program, based on the
optimized geometries mentioned above.

**2.2 Rate coefficients calculations**

The rate coefficients for the hydrolysis of $HNSO_2$ with $CH_3SO_3H$ were calculated through a
two-step process. Initially, the high-pressure-limit (HPL) rate coefficients were computed applying
VRC-VTST methods within the Polyrate package (Chuang et al., 1999). Subsequently, on the basis
of the HPL rate coefficients, the rate coefficients for the hydrolysis of $HNSO_2$ with $CH_3SO_3H$ were
calculated within the temperature range of 212.6-320.0 K and pressures applying the Master
Equation Solver for Multi-Energy Well Reactions (MESMER) program (Glowacki et al., 2012). The
rate coefficients for the barrierless steps transitioning between reactants and pre-reactive complexes
were assessed applying the Inverse Laplace Transform (ILT) method within MESMER calculations,
while the step transitioning between pre-reactive complexes and post-reactive complexes via
transition states were evaluated using the RRKM theory (Mai et al., 2018) in combination with the
asymmetric Eckart model. The details of the rate coefficient for the hydrolysis of $HNSO_2$ without
and with $X$ ($X$ = $H_2O$ and $CH_3SO_3H$) were given in Part 1, Table 1 and Table S4.

**2.3 BOMD Simulations**

BOMD simulations were conducted applying DFT implemented in CP2K program
(Vandevondele et al., 2005; Hutter et al., 2014). The exchange and correlation interactions were
addressed using the Becke-Lee-Yang-Parr (BLYP) functional (Becke, 1988; Lee et al., 1988),
while Grimme's dispersion was applied to address weak dispersion interaction (Grimme et al.,
2010). The Goedecker-Teter-Hutter (GTH) conservation pseudopotential (Goedecker et al.,
1996; Hartwigsen et al., 1998) combine with Gaussian DZVP basis set (Vandevondele and
Hutter, 2007) and an auxiliary plane wave basis set were used to represent core and valence
electrons. Energy cutoffs (Zhong et al., 2017; Zhong et al., 2018; Zhong et al., 2019) of 280 Ry
for the plane wave basis set and 40 Ry for the Gaussian basis set were applied. The gaseous
reactions were simulated in the NVT ensemble at 300 K, with $15 \times 15 \times 15$ Å$^3$ supercells and the



time step of 1 fs. To simulate the water microdroplet, the system containing 191 water molecules
(Zhong et al., 2017) was utilized in $35 \times 35 \times 35$ Å$^3$ supercells. This setup included $HNSO_2$ and
$CH_3SO_3H$ along with the water drop. Prior to the interfacial simulation, a 10 ps relaxation period
in the BOMD simulation was used to equilibrate the water microdroplet system with 191
molecules.

**2.4 ACDC kinetics simulation**

The ACDC model was utilized to simulate the $(MSA)_x(MA)_y(SFA)_z$ ($0 \leq y \leq x + z \leq 3$)
cluster formation rates and explore the potential mechanisms. This simulation encompasses a
variety of temperatures and monomer concentrations to capture the dynamics under different
environmental conditions. Thermodynamic parameters, obtained from quantum chemical
calculations executed at the M06-2X/6-311++G(2$df$,2$pd$) level, were used as inputs for the
ACDC model. The temporal progression of cluster concentrations was determined by
numerically integrating the birth-death equation, leveraging MATLAB's ode15s solver for
enhanced accuracy.
$$\frac{dc_i}{dt} = \frac{1}{2} \sum_{j<i} \beta_{j,(i-j)} c_j c_{(i-j)} + \sum_j \gamma_{(i+j)\to i} c_{i+j} - \sum_j \beta_{i,j} c_i c_j - \frac{1}{2} \sum_{j<i} \gamma_{i\to j} c_i + Q_i - S_i \quad (2)$$
Here, $c_i$ represents the concentration of a specific cluster, labelled as $i$; the term $\beta_{i,j}$ was used to
denote the collision coefficient, which was a measure of the frequency at which clusters $i$ and $j$
collide with each other in a given environment or system; the coefficient $\gamma_{(i+j)\to i}$ was defined
as the evaporation rate constant that describes the process of a larger cluster, consisting of
combined elements $i$ and $j$, breaking down into the individual smaller clusters $i$ and $j$; and $Q_i$
encompasses all other source terms contributing to the formation of cluster $i$. $S_i$ signifies
alternative sink terms for cluster $i$ that may remove it from the system.

**3. Results and discussions**

**3.1 The hydrolysis of $HNSO_2$ with $CH_3SO_3H$ in the gas phase**

Given the low chance of three molecules of $HNSO_2$, $H_2O$ and $CH_3SO_3H$ colliding
simultaneously under atmospheric conditions, the hydrolysis of $HNSO_2$ with $CH_3SO_3H$ (Channel
MSA) was likely a sequential bimolecular process. As the concentration of water molecule ($10^{18}$
molecules·cm$^{-3}$) in the atmosphere is much higher than those of $HNSO_2$ and $CH_3SO_3H$ ($10^5$-$10^9$



molecules·cm$^{-3}$), the reaction pathway of HNSO$_2$···CH$_3$SO$_3$H + H$_2$O is hard to occur in actual
atmospheric conditions. So, Channel MSA proceeds through the initial formation of dimers
(HNSO$_2$···H$_2$O and CH$_3$SO$_3$H···H$_2$O) via collisions between HNSO$_2$ (or CH$_3$SO$_3$H) and H$_2$O.
Subsequently, the generated dimer interacts with the third reactant, either CH$_3$SO$_3$H or HNSO$_2$. As
seen in Fig. 1, the calculated Gibbs free energy of CH$_3$SO$_3$H···H$_2$O complex was -0.9 kcal·mol$^{-1}$,
which was 4.5 kcal·mol$^{-1}$ lower than that of HNSO$_2$···H$_2$O. Consequently, it was predicted the
primary route for the hydrolysis reaction of HNSO$_2$ with CH$_3$SO$_3$H takes place via the HNSO$_2$ +
CH$_3$SO$_3$H···H$_2$O reaction.

Starting from the HNSO$_2$ + CH$_3$SO$_3$H···H$_2$O reactants, the Channel MSA was initiated through

the intermediate complex designated as IM_MSA1. From a geometric perspective, IM_MSA1
complex exhibits a cage-like configuration by a van der Waals force (S1···O1, 2.00 Å) and the
involvement of three hydrogen bonds of H2···O4 (1.53 Å), H4···N1 (1.60 Å) and H5···O3 (2.07 Å).
The Gibbs free energy of IM_MSA1 complex relative to HNSO$_2$ + CH$_3$SO$_3$H···H$_2$O reactants was
1.7 kcal·mol$^{-1}$. Subsequently, as presented in Fig. 1, Channel MSA progresses through transition
state TS_MSA1 to yield complex IMF_MSA1. At TS_MSA1, the CH$_3$SO$_3$H moiety facilitates two
hydrogen atom transfer, with TS_MSA1 lying only 0.8 kcal·mol$^{-1}$ above complex IM_MSA1.
Complex IMF_MSA1 exhibits a cage-like structure with a Gibbs free energy was 23.4 kcal·mol$^{-1}$
lower than that of IM_MSA1, revealing thermodynamic favorability of HNSO$_2$ hydrolysis with
CH$_3$SO$_3$H. To evaluate the relative catalytic impact of CH$_3$SO$_3$H and H$_2$O, Fig. S4 illustrates the
profiles of Gibbs free energy for the hydrolysis of HNSO$_2$ and the corresponding reaction assisted
by H$_2$O. Compared to complex HNSO$_2$···(H$_2$O)$_2$, the Gibbs stabilization energy of IM_MSA1
increased by 5.6 kcal·mol$^{-1}$, potentially shortening the S1···O1 bond distance by 0.21 Å.
Considering the Gibbs free energy barrier and rate coefficients, CH$_3$SO$_3$H demonstrates a greater
catalytic role compared to H$_2$O in lowering the energy barrier for the hydrolysis of HNSO$_2$. In
particular, CH$_3$SO$_3$H facilitates hydrogen atom to extraction from H$_2$O, further reducing the reaction
energy barriers to 7.7 kcal·mol$^{-1}$. Meanwhile, the calculated rate coefficients for HNSO$_2$ hydrolysis
with CH$_3$SO$_3$H was $3.08 \times 10^{-11}$-$3.50 \times 10^{-11}$ cm$^3$·molecule$^{-1}$·s$^{-1}$ within 212.6-320.0 K, exceeding
corresponding values for reactions involving H$_2$O by 2 orders of magnitude. Besides, the Gibbs free
energy of IMF_MSA1 was 2.0 kcal·mol$^{-1}$ lower than that of the product complex IMF_WM1
(NH$_2$SO$_3$H···H$_2$O), suggesting NH$_2$SO$_3$H has a higher affinity for CH$_3$SO$_3$H compared to H$_2$O.



Besides, $CH_3SO_3H$-assisted $HNSO_2$ hydrolysis is reduced by 4.9 kcal·mol$^{-1}$ in energy barrier than
the $NH_3$-assisted ammonolysis of $SO_3$ with their rate constants close each other ($4.35 \times 10^{-10}$
cm$^3$·molecule$^{-1}$·s$^{-1}$) (Li et al., 2018). As the absence of the concentration of $HNSO_2$, the
competitiveness of these two reactions cannot be further confirmed.
To evaluate the comparative catalytic ability of $X$ ($X = H_2O$ and $CH_3SO_3H$) in the atmosphere,
the effective rate coefficients ($k'$) for $X$-assisted $HNSO_2$ hydrolysis were calculated in Table 1.
Notably, $k'$ serves as a metric for gauging the comparative catalytic ability of a series of gaseous
catalysts in atmospheric reactions (Sarkar et al., 2017; Zhang et al., 2020; Zhang et al., 2019; Buszek
et al., 2012; Gonzalez et al., 2011; Parandaman et al., 2018; Anglada et al., 2013). When $X$ was
present, the calculated $k'$ was given by Eq. (3).
$$k'_X = k_X \times K_{eq}(X \cdots H_2O) \times [X] \qquad (3)$$
In Eq. (3), $k_X$ was the rate coefficient for $X$-assisted $HNSO_2$ hydrolysis (Table 1), while $K_{eq}(X \cdots H_2O)$
denotes the equilibrium coefficients of $X \cdots H_2O$ (Table S2). [$X$] represents the avaiable
concentrations of $H_2O$ (Anglada et al., 2013) and $CH_3SO_3H$ (Shen et al., 2020). As indicated in
Table 1, at experimental concentrations ([$H_2O$] = $5.16 \times 10^{16}$-$2.35 \times 10^{18}$ molecules·cm$^{-3}$) within
280.0-320.0 K (at 0 km), the computed $k'_{WM}$ ranged from $5.99 \times 10^{-18}$-$7.79 \times 10^{-17}$ cm$^3$·molecule$^{-1}$·s$^{-1}$.
This range exceeded $k'_{MSA}$ ($4.60 \times 10^{-21}$-$4.81 \times 10^{-20}$ cm$^3$·molecule$^{-1}$·s$^{-1}$) by 2-4 orders of
magnitude, highlighting pronounced impact of $H_2O$ compared to $CH_3SO_3H$ at 0 km in enhancing
the rate of $HNSO_2$ hydrolysis. However, with the significant decrease in atmospheric water
molecules with increasing altitude, the calculated $k'_{MSA}$ ranged from $1.96 \times 10^{-19}$·s$^{-1}$-$1.30 \times 10^{-17}$·cm$^3$·molecule$^{-1}$·s$^{-1}$,
surpassing $k'_{WM}$ ($9.85 \times 10^{-27}$-$6.51 \times 10^{-22}$·cm$^3$·molecule$^{-1}$·s$^{-1}$) by 3-10 orders
of magnitude. This illustrates that $CH_3SO_3H$ has a significantly greater catalytic ability than $H_2O$
in accelerating the rate of $HNSO_2$ hydrolysis within 0-10 km. So, $HNSO_2$ hydrolysis with $CH_3SO_3H$
may represent a potential formation pathway for $NH_2SO_3H$ across an altitude scope of 5-15 km.
**3.2 Reactions at the air-water interface**
The interfacial mechanism of $CH_3SO_3H$-assisted $HNSO_2$ hydrolysis at the air-water
interface has not been thoroughly investigated. Interestingly, our simulations show that $HNSO_2$ and
$CH_3SO_3H$ molecules spend approximately 49.1% and 12.1% of the time, respectively, at the air-
water interface during the 150 ns simulation (Fig. S1 and Fig. S6). This reveals that the presence of
$HNSO_2$ and $CH_3SO_3H$ at the air-water interface should not be disregarded. Therefore, BOMD



simulations were performed to clarify the interfacial mechanism of $CH_3SO_3H$-assisted $HNSO_2$
hydrolysis at the air-water interface. Comparable to the reactions of $SO_3$ at the air-water interface
with acidic molecules (Cheng et al., 2023; Zhong et al., 2019a), the hydrolysis of $HNSO_2$ with
$CH_3SO_3H$ at the air-water interface may occur through three pathways: (*i*) the adsorbed $CH_3SO_3H$
interacts with $HNSO_2$ at the air-water interface; (*ii*) the adsorbed $HNSO_2$ interacts with $CH_3SO_3H$
at the air-water interface; and (*iii*) the $HNSO_2{\cdots}CH_3SO_3H$ complex reacts at the air-water interface.
Nevertheless, because of the high reactiveness of $CH_3SO_3H$ at the air-water interface, the lifetime
of $CH_3SO_3H$ was minimal (seen in Fig. S9) on the water droplet, which was around a small number
of picoseconds leading to the rapid formation of $CH_3SO_3^-$ ion. Meanwhile, although $HNSO_2$
remains stable at the air-water interface (seen in Fig. S8) and does not dissociate within 10 ps, the
hydrated form of $HNSO_2$ illustrated in Fig. S8 was not conducive to $HNSO_2$ hydrolysis at the air-
water interface. So, model (*iii*) was primarily considered for $HNSO_2$ hydrolysis with $CH_3SO_3H$ at
the air-water interface. It was worth noting that $HNSO_2{\cdots}CH_3SO_3H$ complex can persist at the air-
water interface for approximately 34.2% of the 150 ns simulation time (see in Fig. S7). For model
(*iii*), two types of reactions were found at the air-water interface: (*a*) the $NH_2SO_3^-$ and $H_3O^+$ ions
formation mechanism, and (*b*) the proton exchange mechanism.
**$NH_2SO_3^-$ and $H_3O^+$ ions forming mechanism.** Fig. 2(a), Fig. S10 and Movie 1 illustrates the
formation mechanism of $NH_2SO_3^-$ and $H_3O^+$ ions through the chain structure. At 4.57 ps, a chain
hydrolyzed transition state was observed, accompanied by two protons transfer events. Specially,
an H2 atom transferred from the OH moiety of $CH_3SO_3H$ molecule to the terminal N atom of
$HNSO_2$ molecule, resulting in the breaking of the O3-H2 bond (with the length of 1.49 Å) and
forming the H2-N bond (with the length of 1.14 Å). Concurrently, an interfacial water molecule
decomposes, leading to the elongation of the O1-H1 bond to over 1.00 Å, with the S1 atom of
$HNSO_2$ obtaining the OH moiety of the interfacial water molecule ($d_{(S1\text{-}O1)}$ = 1.60 Å). By 4.61 ps,
The N-H2 and S1-O1 bonds both shortened to 0.99 Å and 1.01 Å, revealing the formation of the
$NH_2SO_3H$ molecule. However, due to its strong acidity, the $NH_2SO_3H$ molecule could only persist
on the water droplet surface for a ps time-scale. As a result, at 7.43 ps, the proton of $NH_2SO_3H$
transferred to another interfacial water molecule, completing the deprotonation of $NH_2SO_3H$. The
loop structure mechanism (Fig. 2(b), Fig. S11 and Movie 2) was similar with the chain structure
mechanism. However, in this case, the proton of $NH_2SO_3H$ transferred to $CH_3SO_3^-$ rather than to an



interfacial water molecule.
**Proton exchange mechanism**. As depicted in Fig. 3, the proton exchange mechanism
illustrates the deprotonation of $CH_3SO_3H$ concurrent with $HNSO_2$ hydration at the air-water
interface. As shown in Fig. 3(a), Fig. S12 and Movie 3, $CH_3SO_3H$-mediated hydration $HNSO_2$ with
a single water molecule was observed. Initially, the $HNSO_2\cdots CH_3SO_3H$ complex quickly associates
with an interfacial water molecule, and forms a loop structure complex that accelerates the rate of
proton transfer. By 4.38 ps, an eight-membered loop structure complex, $HNSO_2\cdots H_2O\cdots CH_3SO_3H$,
emerges, characterized by two hydrogen bonds ($d_{(H2-N)}$ = 1.82 Å and $d_{(H1-O2)}$ = 1.92 Å) and a van
der Waals forces ($d_{(S1-O1)}$ = 2.35 Å). Thereafter, at 4.77 ps, a transition state-like configuration was
identified where the water molecule within the loop complex dissociated, elongating the O1-H1
bond to over 1.00 Å, and the S atom of $HNSO_2$ attaches to the OH group of the interfacial water
molecule. Concurrently, the $CH_3SO_3^-$ ion receives the proton from the separated interfacial water
molecule. The entire reaction for $CH_3SO_3H$-mediated hydration $HNSO_2$ with one water molecule
was completed at 4.80 ps, resulting in the formation of $NH_2SO_3H$ and $CH_3SO_3H$ molecules.
$CH_3SO_3H$-mediated hydration of $HNSO_2$ with two water molecules (Fig. 3(b), Fig. S13 and Movie
4) at the air-water interface was similar with mechanism identified with one water molecule.
However, the inclusion of two water molecules enlarges the loop, significantly reducing the stress
on the loop structures. Consistent with the prediction in Fig. 4, the loop structures preferred to
include two water molecules rather than one water molecule. This observation agrees well with the
reported hydration of Criegee intermediate at the air-water interface (Zhu et al., 2016; Kumar et al.,
2018; Liu et al., 2021; Zhang et al., 2023a). Additionally, $CH_3SO_3H$-mediated hydration of $HNSO_2$
with three water molecules (Fig. S14 and Movie 5) has been observed in the proton exchange
mechanism. However, its probability of occurrence was smaller due to the relatively larger entropy
effect. It was noteworthy that the $NH_2SO_3H$ and $CH_3SO_3H$ molecules formed in the proton exchange
mechanism were not stable at the air-water interface, which can further interact with an interfacial
water molecule to form the corresponding ions of $NH_2SO_3^-$ and $CH_3SO_3^-$.
At the air-water interface, a sum of 50 BOMD trajectories, each lasting 10 ps, were conducted
to investigate $HNSO_2$ hydrolysis with $CH_3SO_3H$. Two distinct mechanisms were observed: the
formation of $NH_2SO_3^-$ and $H_3O^+$ ions formation (shown in blue and yellow in Fig. 4) and the proton
exchange mechanism (represented by orange, purple and green in Fig. 4). In the mechanism



involving the formation of $NH_2SO_3^-$ and $H_3O^+$ ions, approximately 22% (Fig. 2(a), Fig. 4, Fig. S10
and Movie 1) of the reactions took place via a chain structure, while the majority (~18%) (Fig. 2(b),
Fig. 4, Fig. S11 and Movie 2) proceeded through a loop structure mechanism. This discrepancy can
be attributed to the uncertainty regarding the direction of proton transfer from $NH_2SO_3H$. Since the
number of water molecules near the water microdroplet far exceeded that of $CH_3SO_3^-$, protons were
predominantly transferred to interface water molecules, making the loop structure mechanism
weaker than the chain structure mechanism. Approximately 60% of the reactions were observed to
be due to the proton exchange mechanism in BOMD simulations. Through water-mediated
mechanisms, these reactions resulted in $NH_2SO_3H$ formation. Similarly to gas-phase reactions, loop
structures were observed in these reactions. Approximately 10% of the reactions formed a loop
structure involving one water molecule (Fig. 3(a), Fig. 4, Fig. S12 and Movie 3), while the most
common loop structure involved two water molecules (about 42%) (Fig. 3(b), Fig. 4, Fig. S13 and
Movie 4). Smaller loops were found to experience more stress than loop structures with two water
molecules. In cases of loop structures with three water molecules (about 8%) (Fig. 4, Fig. S14 and
Movie 5), the entropy effect was deemed to be more significant than the strain effect and likely
played a dominant role. The two water molecules contained in the loop structure not only acted as
a reactant but also facilitated proton transfer as a bridge.
**3.3 New Particle Formation from the atmospheric products**
**3.3.1. The influence of SFA on the stability of atmospheric MSA-MA-based**
**clusters**
Electrostatic Potential (ESP) mapping on the molecular van der Waals (vdW) surface was
employed to analyze the interactions between SFA and other key nucleation precursors like MSA
and MA. As shown in Fig. 5, sites with more negative ESP often attract more positive ESP sites,
namely hydrogen bonds in the studied system. Specifically, the hydrogen atoms of the $-SO_3H$ and -
$NH_2$ groups (site 4 and 5) in SFA, possessing more positive ESP values, have the potential to attract
groups with negative ESP values, such as the oxygen atom within the $-SO_3H$ group of MSA (site 6)
and the nitrogen atom of MA (site 1), thus forming hydrogen bonds as proton donors. Additionally,
the sulfur atom of the $-SO_3H$ functional group (site 7) in SFA, with a negative ESP of -30.75, acts
as proton acceptor, facilitating direct binding with MSA and MA molecules via the hydrogen bonds.



326 Therefore, the introduction of SFA was believed to enhance the stability of MSA-MA clusters by

327 promoting the formation of more hydrogen bonds and facilitating proton transfers.

### 3.3.2. The cluster formation rates in the SFA-MSA-MA system

329 Simulations were conducted to determine the cluster formation rates ($J$) for the MSA-MA-SFA

330 system, varying parameters such as temperature and the concentrations of the precursors were

331 involved. To assess the promotional impact of SFA on $J$ under varying atmospheric conditions, the

332 enhancement factor ($R$) was computed as the ratio of $J_{MSA-MA-SFA}$ to $J_{MSA-MA}$. As depicted in Fig. 6

333 (a), the $J$ of MSA-MA-SFA system exhibits a negative correlation with temperature, attributed to

334 the decrease in $\Delta G$ value and evaporation rates of clusters at lower temperatures. Conversely, a

335 positive correlation of $R$ with temperature was observed (Fig. 6(b)), indicating that SFA's

336 enhancement of nucleation was more pronounced in regions with relatively higher temperatures.

337 Furthermore, both $J$ and $R$ show an increase as the [SFA] increases, suggesting a positive correlation

338 of $J$ and $R$ with [SFA]. In short, in regions with high [SFA], such as the Yangtze River Delta of

339 China, Bangladesh, and the east coast of India, SFA was expected to significantly boost the $J$ of

340 MSA-MA based nucleation. It is noted that in Fig. 6(b), due to the competitive relationship between

341 MSA and SFA, at low concentrations of SFA, the binding capacity of MSA with MA is stronger

342 than that of SFA with MA, resulting in only a small amount of SFA participating in cluster formation.

343 However, as the concentration of SFA increases, the number of $(MSA)_x \cdot (MA)_y \cdot (SFA)_z$ (where $y \leq$

344 $x + z \leq 3$) ternary clusters increases, leading to the formation of more hydrogen bonds and a

345 significant increase in $R_{SFA}$. Additionally, Fig. 7 illustrates the $J$ and $R$ of MSA-MA-SFA clusters

346 under different [MSA] and [MA]. On one hand, larger values of [MSA] and [MA] correspond to

347 higher $J$, as the increased concentration of nucleation precursors leads to a rise in the number of

348 MSA-MA-FSA clusters. On the other hand, increasing [MSA] and [MA] result in a decrease in the

349 $R$ attributed to the effect of SFA on nucleation. This was because as [MSA] and [MA] increases,

350 the prevalence of pure MSA-MA clusters rise during the clustering process, consequently reducing

351 the impact of SFA.

### 3.3.3. The growth paths of cluster under different atmospheric conditions

353 In Fig. 8 (a), two main types of cluster formation routes were found: (*i*) the pure MSA-MA

354 pathway and (*ii*) the MSA-MA-SFA pathways at 278.15 K in the studied system. In the pure MSA-

355 MA pathway, cluster growth primarily occurs through the collisional addition of MSA or MA

356 monomers. Conversely, in the SFA-involved pathways, SFA can directly participate in the



formation of stable larger clusters subsequently, such as $(MSA)_2·(MA)_2·SFA$ and
$(MSA)_2·(MA)_2·(SFA)_2$ clusters, and then subsequently grow out. The involvement of SFA in the
cluster formation pathway was significantly influenced by atmospheric conditions. Firstly, as the
temperature rises from 238.15 K to 278.15 K, the contribution of the SFA-involved cluster
formation pathways rises from 68% to 90% (Fig. 8 (b)), implying that the pathway involving SFA
becomes increasingly important at lower altitudes or in warmer conditions. Secondly, the
contribution of the pathway with SFA exhibits a negative correlation with [SA] (Fig. 8 (c)),
attributed to the competitive relationship between SFA and MSA. Thirdly, the contribution of the
SFA-involved cluster formation pathway was positively associated with the concentration of [SFA]
(Fig. 8 (d)). At $[SFA]=10^4$ molecules·$cm^{-3}$, the pathway involving SFA was not prominent, and the
pure MSA-MA pathway dominates. However, as [SFA] rises from $10^4$ to $10^8$ molecules·$cm^{-3}$, the
contribution of the SFA-involved pathway increased from 7% to 98% at 278.15 K. Therefore, the
pathway involving SFA appears to dominate near SFA release sources in warm temperatures or at
lower altitudes.
**3.4 Interfacial implications of products on aerosol particle growth**
As the discussion above, the formation of $SFA^-···H_3O^+$ and $MSA^-···H_3O^+$ ions pairs can occur
within a few picoseconds at the air-water interface. The atmospheric affinity of $MSA^-$, $SFA^-$ and
$H_3O^+$ for gaseous precursors was further probed by evaluating the free energies of interaction. It
was worth noting that compounds such as MSA, MA, $HNO_3$ (NA), and $(COOH)_2$ (OA) were
identified as candidate species for consideration (Wang et al., 2024; Kulmala et al., 2004). As
presented in Table 2, the computed binding energies demonstrate that the interactions of $SFA^-$
···MSA, $SFA^-$···NA, $SFA^-$···OA, $H_3O^+$···MA, $MSA^-$···MSA, $MSA^-$···OA, and $MSA^-$···NA were
stronger than those of MSA···MA (one of the primary precursors for atmospheric aerosols), with
their Gibbs free energies increased by 14.3-50.9 kcal·$mol^{-1}$. The findings indicate that the presence
of $SFA^-$, $MSA^-$, and $H_3O^+$ at the interface facilitates the capture of potential gaseous species onto
the surface of water microdroplet.
Furthermore, we investigated the possibility of $SFA^-$ contributing to the enlargement of
particles within the MSA-MA cluster, taking into account the geometric configuration and the free
energy of formation for the $(MSA)_1·(MA)_1·(SFA^-)_1$ clusters aggregating. Compared with other
clusters, such as $(MSA)_1·(MA)_1·(X)_1$ (where $X$ = HCOOH, $CH_3COOH$, CHOCOOH, OA,
$CH_3COCOOH$, $HOOCCH_2COOH$, $HOOC(CH)_2COOH$, $HOOC(CH_2)_2COOH$,



HOOC(CH$_2$)$_3$COOH, C$_6$H$_5$(COOH) and C$_{10}$H$_{16}$O$_3$) clusters (Zhang et al, 2022), the quantity of
hydrogen bonds within the (MSA)$_1$·(MA)$_1$·(SFA$^-$)$_1$ cluster has increased, and the loop of complex
was expanded. It has been demonstrated that SFA$^-$ has the greatest capacity to stabilize MSA-MA
clusters and facilitate MSA-MA nucleation in these clusters. This was attributed to its acidic nature
and structural characteristics, which include a greater number of intermolecular hydrogen bond
binding sites. Therefore, relative to (MSA)$_1$·(MA)$_1$·($X$)$_1$ cluster (Table 2), the Gibbs formation free
energy $\Delta G$ of the (MSA)$_1$·(MA)$_1$·(SFA$^-$)$_1$ cluster was lower, indicating that the NH$_2$SO$_3^-$ ion
exhibits a more potent nucleation capacity at the air-water interface compared to the $X$ species in
the gas phase. Consequently, our forecast was that the presence of NH$_2$SO$_3^-$ at the air-water interface
will foster enhanced particle growth.

## 398    4. Summary and Conclusions

In this study, quantum chemical calculations, BOMD simulations and ACDC kinetic model

were utilized to characterize the gaseous and interfacial hydrolysis of HNSO$_2$ with CH$_3$SO$_3$H, and
to examine the influence exerted by NH$_2$SO$_3$H on MSA-MA-based clusters.

In the gaseous reaction, the activation energy for the hydrolysis of HNSO$_2$ catalyzed by

CH$_3$SO$_3$H was only 0.8 kcal·mol$^{-1}$, significantly lower by 7.7 kcal·mol$^{-1}$ than the energy barrier of
H$_2$O-assisted HNSO$_2$ hydrolysis. The effective rate coefficients reveal that the NH$_2$SO$_3$H formation
from CH$_3$SO$_3$H-catalyzed hydrolysis of HNSO$_2$ can be competitive with that catalyzed by H$_2$O
within an altitude of 5-15 km. Moreover, kinetic simulations utilizing the ACDC have disclosed that
SFA has an unexpectedly positive impact on the NPF process, markedly enhancing the assembly of
MSA-MA-based cluster. Notably, the "participant" mechanism of SFA for cluster formation has
been identified by tracing the growth paths of the system in agriculture-developed and coastal
industrial areas, especially significant in the Yangtze River Delta of China, Bangladesh, and the east
coast of India.

At the air-water interface, the NH$_2$SO$_3^-$ and H$_3$O$^+$ ions forming mechanism (~40%) and the

proton exchange mechanism (~60%) were observed in the hydrolysis of HNSO$_2$ with CH$_3$SO$_3$H,
which can take place in a few picoseconds. Notably, the formed NH$_2$SO$_3^-$, CH$_3$SO$_3^-$, and H$_3$O$^+$ ions
at the air-water interface possess the ability to attract potential precursor molecules like CH$_3$SO$_3$H,
CH$_3$NH$_2$, and HNO$_3$. This attraction facilitates the transition of gaseous molecules onto the surface



of water microdroplet. Moreover, the assessment of the potential of $X$ in the formation of the ternary
$CH_3SO_3H$-$CH_3NH_2$-$X$ cluster revealed that $NH_2SO_3^-$ exhibits the greatest propensity to stabilize
$CH_3SO_3H$-$CH_3NH_2$ clusters and to foster nucleation of $CH_3SO_3H$-$CH_3NH_2$ in the context of $X$.

Overall, this work not only elucidates a novel mechanism underlying the hydrolysis of $HNSO_2$

with $CH_3SO_3H$, but also highlight the potential contribution of SFA on aerosol particle growth and
new particle formation.
**Acknowledgments**

This work was supported by the National Natural Science Foundation of China (No: 22203052;

22073059); the Key Cultivation Project of Shaanxi University of Technology (No: SLG2101); the
Education Department of Shaanxi Provincial Government (No. 23JC023).
**Declaration of competing interest**

The authors declare that they have no known competing financial interests or personal

relationships that could have appeared to influence the work reported in this paper.



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



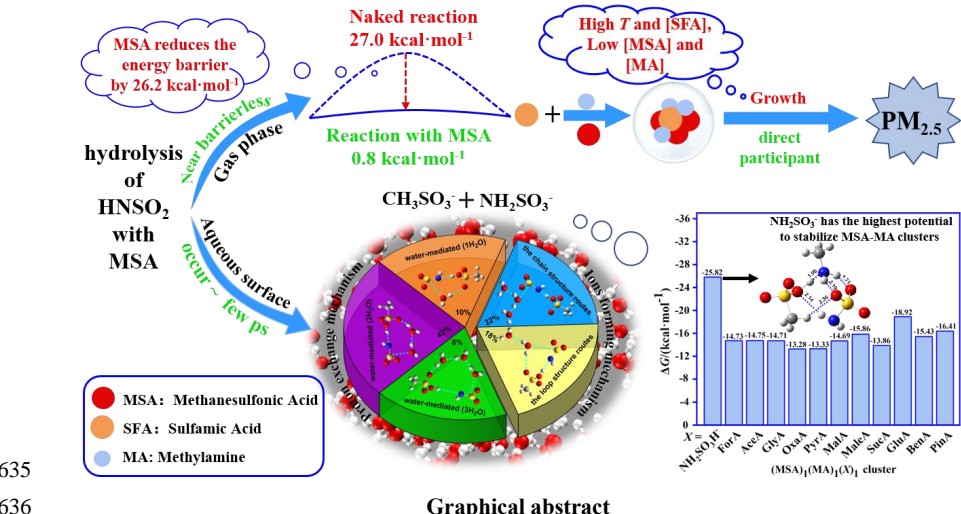


**Graphical abstract**





**Figure Captions**


**Fig. 1** The potential energy profile ($\Delta G$) for the hydrolysis reaction of $HNSO_2$ with $CH_3SO_3H$ at the
CCSD(T)-F12/cc-pVDZ-F12//M06-2X/6-311++G($2df$,$2pd$) level of theory


**Fig. 2** BOMD simulation trajectories and snapshots of $NH_2SO_3^-$ and $H_3O^+$ ions forming mechanism
(chain structure (a) and loop-structure (b)) in the $HNSO_2$ hydrolysis with $CH_3SO_3H$ at the air-water
interface


**Fig. 3** BOMD simulation trajectories and snapshots of proton exchange mechanism in $CH_3SO_3H$-
mediated hydration $HNSO_2$ with one (a) and two (b) water molecules at the air-water interface


**Fig. 4** Percentages of different mechanisms for the $HNSO_2$ hydrolysis with $CH_3SO_3H$ at the air-
water interface observed in BOMD simulations


**Fig. 5** ESP-mapped molecular vdW surface of MA, SFA and MSA molecules at M06-2X/6-
311++G($2df$,$2pd$) level of theory. Surface local minima and maxima of ESP of the different
functional groups in MA, SFA and MSA molecules are represented as blue and yellow spheres,
respectively. The values of maximum and minimum are shown in kcal mol$^{-1}$ in the parentheses. The
green, red and blue arrows refer to the tendencies to form hydrogen bonds and proton transfer events,
respectively. (green = carbon, red = oxygen, blue = nitrogen, yellow = sulfur and white = hydrogen.)


**Fig. 6** The $J$ (cm$^{-3}$ s$^{-1}$) (a) and $R$ (b) versus [SFA] with [MSA] = $10^6$ molecules cm$^{-3}$, [MA] = 2.5 ×
$10^8$ molecules cm$^{-3}$ and four different temperatures (green line: 298.15 K, blue line: 278.15 K, red
line: 258.15 K, black line: 238.15 K).


**Fig. 7** The $J$ (cm$^{-3}$ s$^{-1}$) (a) and $R$ (b) as a function of [MSA] with [SFA] = $10^8$ molecules cm$^{-3}$ and
three different [MA] (black line: [MA] = 2.5 × $10^7$ molecules cm$^{-3}$, red line: [MA] = 2.5 × $10^8$
molecules cm$^{-3}$, blue line: [MA] = 2.5 × $10^9$ molecules cm$^{-3}$) at 278.15 K.


**Fig. 8** Main cluster formation mechanism of MSA-MA-SFA-based system at 278.15 K, [MSA] =
$10^7$ molecules·cm$^{-3}$, [MA] = 2.5 × $10^8$ molecules·cm$^{-3}$, and [SFA] = $10^6$ molecules·cm$^{-3}$. (a) The
black arrows indicate the pure MSA-MA-based growth pathways. Blue arrows represent the
pathways containing SFA. The influence of (b) temperature, (c) [SFA] and (d) [MSA] on the relative
contribution of the pure MSA-MA-based clustering pathway and the SFA participation pathway to
the system flux is analyzed. Others in (b), (c), and (d) indicate that the pathway contribution of the
cluster growing out of the studied system is less than 5%



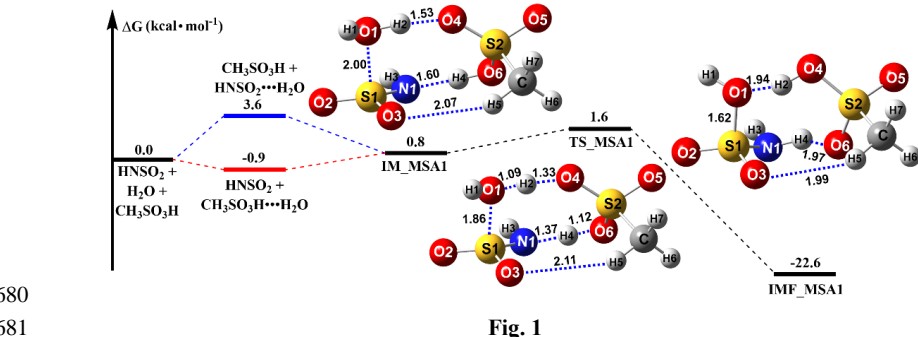


**Fig. 1**



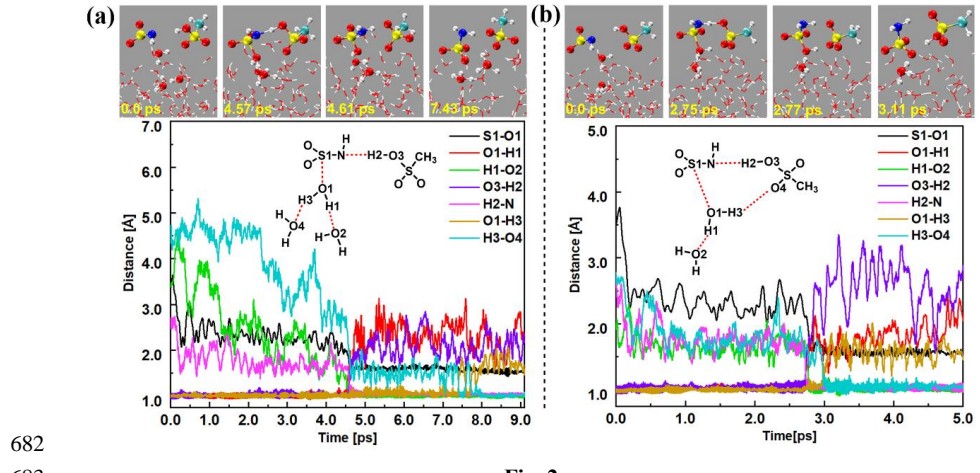


**Fig. 2**



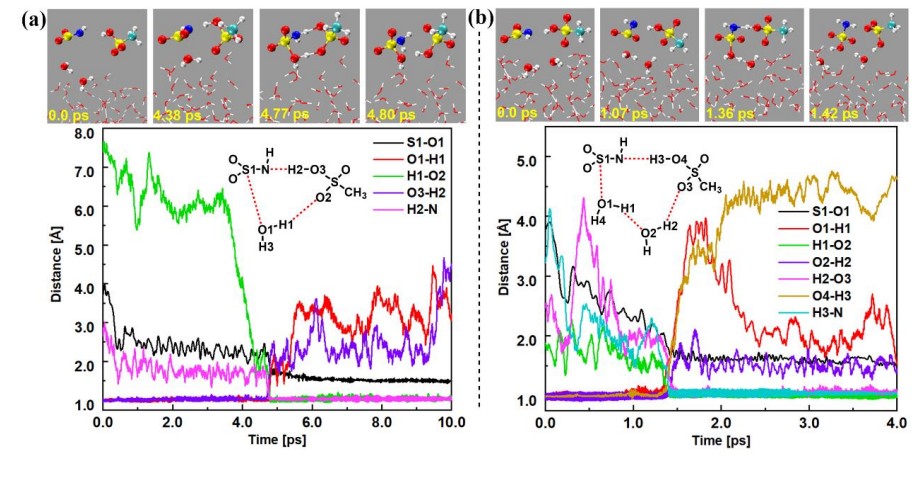



**Fig. 3**



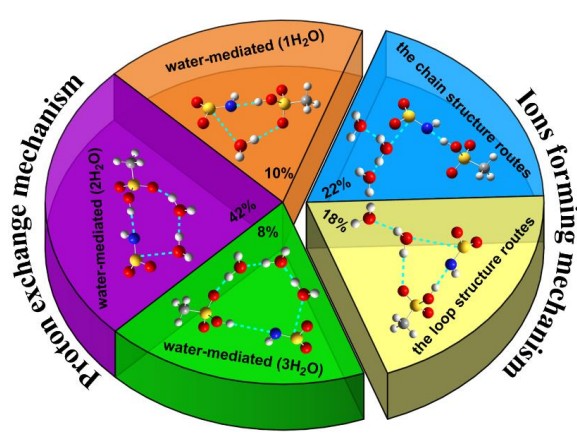


**Fig. 4**





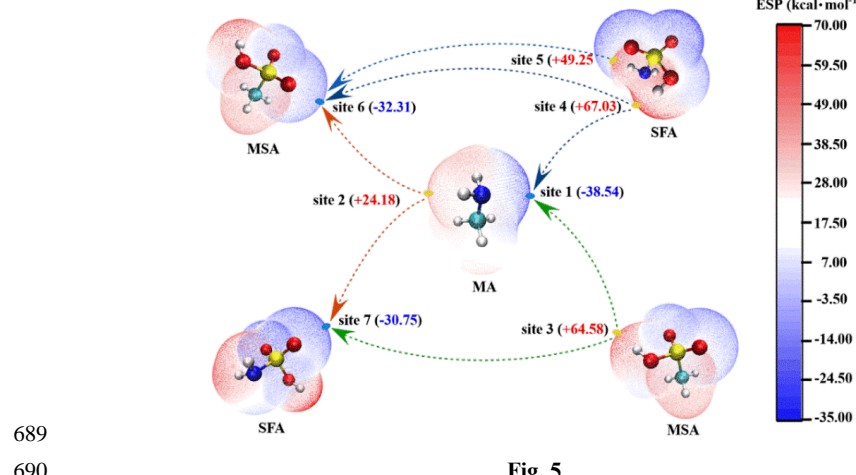


**Fig. 5**





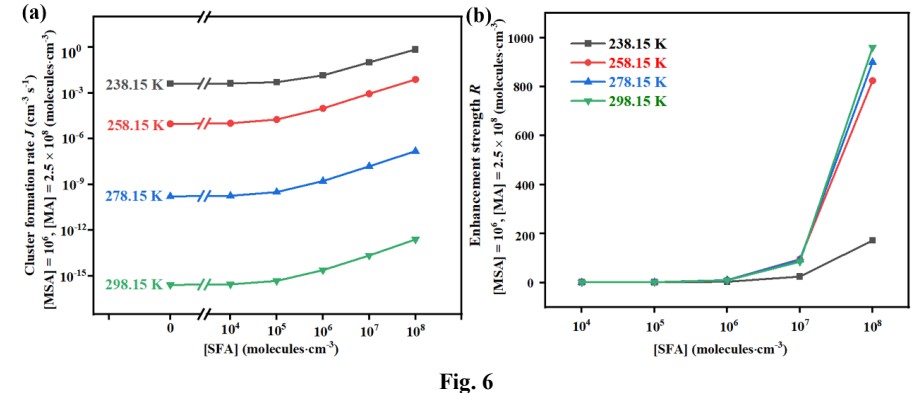

**Fig. 6**





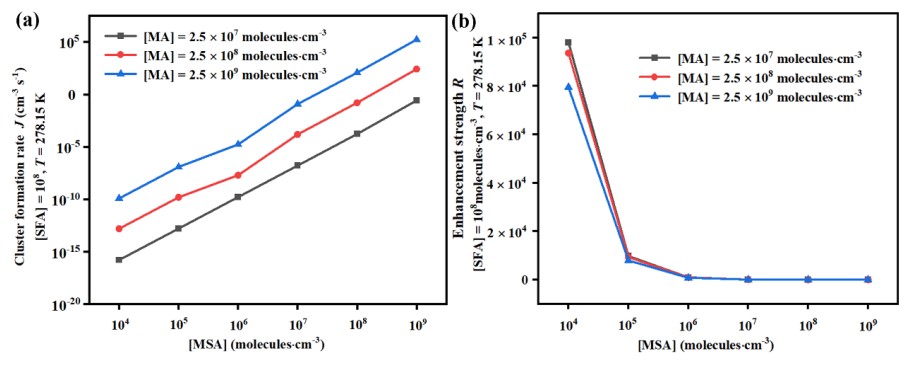


**Fig. 7**



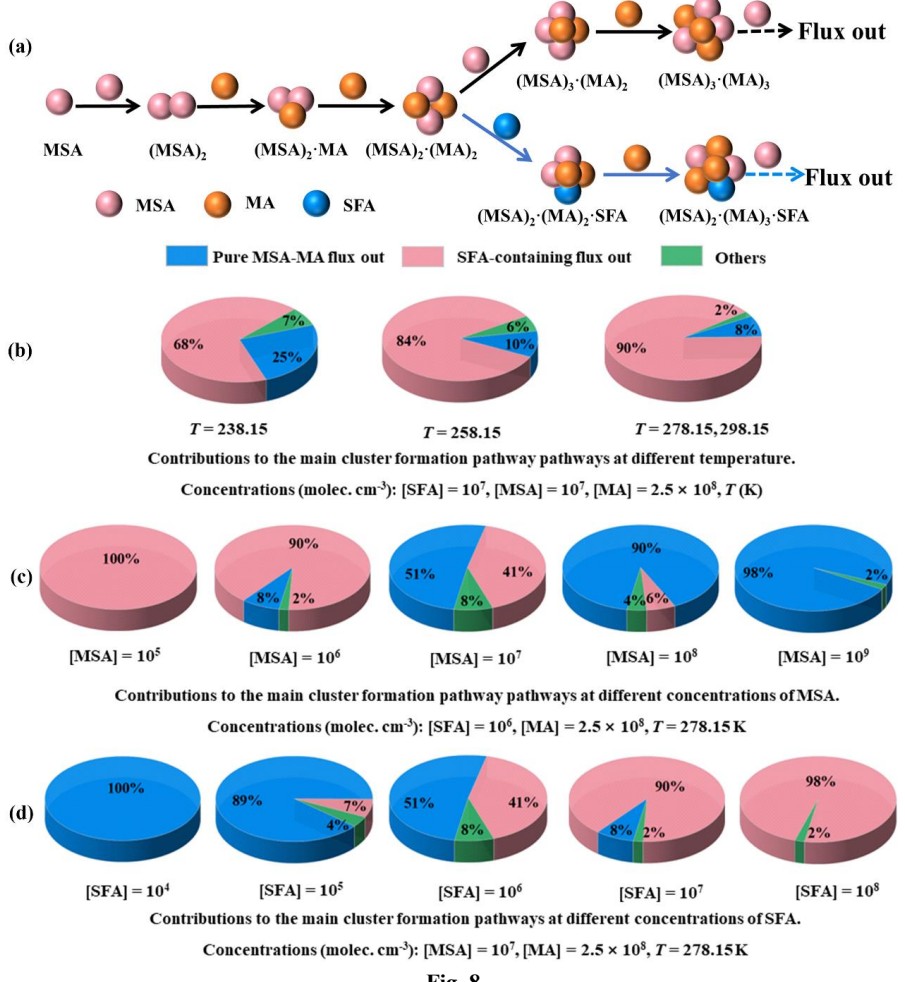

Contributions to the main cluster formation pathway pathways at different temperature.

Concentrations (molec. cm$^{-3}$): [SFA] = 10$^7$, [MSA] = 10$^7$, [MA] = 2.5 × 10$^8$, $T$ (K)

Contributions to the main cluster formation pathway pathways at different concentrations of MSA.

Concentrations (molec. cm$^{-3}$): [SFA] = 10$^6$, [MA] = 2.5 × 10$^8$, $T$ = 278.15 K

Contributions to the main cluster formation pathways at different concentrations of SFA.

Concentrations (molec. cm$^{-3}$): [MSA] = 10$^7$, [MA] = 2.5 × 10$^8$, $T$ = 278.15 K

**Fig. 8**





**Table 1** Rate coefficients ($k$, $cm^3 \cdot molecule^{-1} \cdot s^{-1}$) and effective rate constants ($k'$, $cm^3 \cdot molecule^{-1} \cdot s^{-1}$) for the hydrolysis of $HNSO_2$ with $H_2O$ and $CH_3SO_3H$ calculated by master equation within the temperature range of 213-320 K and altitude range of 0-15 km

| Altitude | | 0 km | | | | | | 5 km | 10 km | 15 km |
|---|---|---|---|---|---|---|---|---|---|---|
| T/K | | 280 | 290 | 298 | 300 | 310 | 320 | 259.3 | 229.7 | 212.6 |
| $k_{WM}$ | | $7.64 \times 10^{-13}$ | $6.45 \times 10^{-13}$ | $5.63 \times 10^{-13}$ | $5.44 \times 10^{-13}$ | $4.59 \times 10^{-13}$ | $3.88 \times 10^{-13}$ | $1.09 \times 10^{-12}$ | $1.72 \times 10^{-12}$ | $2.22 \times 10^{-12}$ |
| $k_{MSA}$ | | $3.08 \times 10^{-11}$ | $2.96 \times 10^{-11}$ | $2.85 \times 10^{-11}$ | $2.82 \times 10^{-11}$ | $2.67 \times 10^{-11}$ | $2.52 \times 10^{-11}$ | $3.32 \times 10^{-11}$ | $3.49 \times 10^{-11}$ | $3.50 \times 10^{-11}$ |
| $k'_{WM}$ | 20%RH | $5.99 \times 10^{-18}$ | $7.96 \times 10^{-18}$ | $9.64 \times 10^{-18}$ | $1.03 \times 10^{-17}$ | $1.29 \times 10^{-17}$ | $1.36 \times 10^{-17}$ | | | |
| | 40%RH | $1.19 \times 10^{-17}$ | $1.58 \times 10^{-17}$ | $1.99 \times 10^{-17}$ | $2.07 \times 10^{-17}$ | $2.60 \times 10^{-17}$ | $3.12 \times 10^{-17}$ | | | |
| | 60%RH | $1.79 \times 10^{-17}$ | $2.38 \times 10^{-17}$ | $2.98 \times 10^{-17}$ | $3.11 \times 10^{-17}$ | $3.90 \times 10^{-17}$ | $4.68 \times 10^{-17}$ | $9.85 \times 10^{-27}$ | $1.71 \times 10^{-22}$ | $6.51 \times 10^{-22}$ |
| | 80%RH | $2.39 \times 10^{-17}$ | $3.17 \times 10^{-17}$ | $3.97 \times 10^{-17}$ | $4.14 \times 10^{-17}$ | $5.21 \times 10^{-17}$ | $6.24 \times 10^{-17}$ | | | |
| | 100%RH | $2.97 \times 10^{-17}$ | $3.96 \times 10^{-17}$ | $4.97 \times 10^{-17}$ | $5.18 \times 10^{-17}$ | $6.50 \times 10^{-17}$ | $7.79 \times 10^{-17}$ | | | |
| $k'_{MSA}$ | $[MSA]=10^8$ | $4.81 \times 10^{-19}$ | $2.50 \times 10^{-19}$ | $1.57 \times 10^{-19}$ | $1.40 \times 10^{-19}$ | $7.90 \times 10^{-20}$ | $4.60 \times 10^{-20}$ | $1.96 \times 10^{-20}$ | $2.37 \times 10^{-19}$ | $1.30 \times 10^{-18}$ |
| $k'_{MSA}/k'_{WM}$ | | $1.62 \times 10^{-4}$ | $6.42 \times 10^{-5}$ | $3.16 \times 10^{-5}$ | $2.69 \times 10^{-5}$ | $1.22 \times 10^{-5}$ | $5.90 \times 10^{-5}$ | $3.01 \times 10^{1}$ | $1.38 \times 10^{3}$ | $1.32 \times 10^{8}$ |

$k_{WM}$ and $k_{MSA}$ are respectively the rate constant for the hydrolysis of $HNSO_2$ with $H_2O$ and $CH_3SO_3H$; $k'_{WM}$ and $k'_{MSA}$ are respectively the effective rate constant for the hydrolysis of $HNSO_2$ with $H_2O$ and $CH_3SO_3H$.



**Table 2.** Gibbs free energy ($\Delta G$) for the formation of SFA$^-$···MSA, SFA$^-$···NA, SFA$^-$···OA, $H_3O^+$···MA, MSA$^-$···MSA, MSA$^-$···OA, and MSA$^-$···NA, MSA···MA, $(MSA)_1 \cdot (MA)_1 \cdot (X)_1$ at 298 K

| | SFA$^-$···MSA | SFA$^-$···HNO$_3$ | SFA$^-$···OA | MSA$^-$···MSA | MSA$^-$···NA |
|---|---|---|---|---|---|
| $\Delta G$ | -23.8 | -21.5 | -25.2 | -23.9 | -22.6 |
| | MSA$^-$··· OA | MSA···H$_3$O$^+$ | | MA···H$_3$O$^+$ | MSA···MA |
| $\Delta G$ | -25.8 | -35.8 | | -57.9 | -7.0 (-7.2)[b] |
| | HCOOH ···MSA···MA | CH$_3$COOH ···MSA···MA | CHOCOOH ···MSA···MA | | OA ···MSA···MA |
| $\Delta G$ | -14.7 (-15.8)[a] | -14.8 (-14.3)[a] | -14.7 (-15.6)[a] | | -13.3 (-12.7)[a] |
| | CH$_3$COCOOH ···MSA···MA | HOOCCH$_2$COOH ···MSA···MA | HOOC(CH)$_2$COOH ···MSA···MA | | HOOC(CH$_2$)$_2$COOH ···MSA···MA |
| $\Delta G$ | -13.3 (-13.0)[a] | -14.7 (-16.7)[a] | -15.9 (-15.3)[a] | | -13.9 (-14.3)[a] |
| | HOOC(CH$_2$)$_3$COOH ···MSA···MA | C$_6$H$_5$(COOH) ···MSA···MA | C$_{10}$H$_{16}$O$_3$ ···MSA···MA | | SFA$^-$ ···MSA···MA |
| $\Delta G$ | -18.9 (-17.9)[a] | -15.4 (-15.3)[a] | -16.4 (-15.3)[a] | | -25.8 |

[a] The value was taken from reference (Zhang, R., Shen, J., Xie, H. B., Chen, J., and Elm, J.: The role of organic acids in new particle formation from methanesulfonic acid and methylamine, Atmos. Chem. Phys., 22, 2639-2650, 10.5194/acp-22-2639-2022, 2022b.)

[b] The value was taken from reference (Zhong, J., Li, H., Kumar, M., Liu, J., Liu, L., Zhang, X., Zeng, X. C., and Francisco, J. S.: Mechanistic Insight into the Reaction of Organic Acids with SO$_3$ at the Air–Water Interface, Angew. Chem. Int. Ed., 131, 8439-8443, 2019.)