# Peer review of "A novel formation mechanism of sulfamic acid and its enhancing"

_EGUsphere, 2024_

## Referee Comment (RC1)

Wang et al. present a novel formation mechanism of sulfamic acid (NH₂SO₃H) and its enhancement effect in methanesulfonic acid-methylamine (MSA-MA) aerosol particle formation. The study centers on the production, consumption, and potential pollution impacts of sulfamic acid over agriculture-intensive and coastal industrial regions. The most part of this manuscript is well written and of broad interest to the readership of *Atmospheric Chemistry and Physics*. I recommend publication in *Atmospheric Chemistry and Physics* after the following comments have been addressed.

**Specific Comments:**

**Comment 1: Pages 2- 3 lines 57-62**: "As the direct hydrolysis of $HNSO_2$ with a high energy barrier takes place hardly in the gas phase, the addition of a second water molecule, formic acid and sulfuric acid ($H_2SO_4$, SA) have been proved to promote the product of $NH_2SO_3H$ through the hydrolysis of $HNSO_2$. However, to the best of our knowledge, the gaseous hydrolysis of $HNSO_2$ with $CH_3SO_3H$ has not yet been investigated"

The necessity for studying the gaseous hydrolysis of $HNSO_2$ with $CH_3SO_3H$ is not sufficiently clarified. Is there any research or evidence indicating that the reaction processes you introduced earlier are insufficient to explain the source of sulfamic acid? If so, please provide additional information.

**Comment 2 Page 6 lines 155-156**: "The ACDC model was utilized to simulate the $(MSA)_x(MA)_y(SFA)_z$ ($0 \leq y \leq x + z \leq 3$) cluster formation rates and explore the potential mechanisms"

The structural stability of clusters directly impacts the nucleation ability of a multi-components system. How was the most stable structure of $(MSA)_x(MA)_y(SFA)_z$ ($0 \leq y \leq x + z \leq 3$) clusters used in this paper obtained?

**Comment 3 Page 6 lines 158-160**: "Thermodynamic parameters, obtained from quantum chemical calculations executed at the M06-2X/6-311++G(2df,2pd) level, were used as inputs for the ACDC model"

Please further justify for why the M06-2X/6-311++G(2df,2pd) level of theory was employed to obtain the thermodynamic parameters used as inputs for the ACDC model.

**Comment 4 Page 13 lines 362-366**: "Secondly, the contribution of the pathway with SFA exhibits a negative correlation with [SA] (Fig. 8 (c)), attributed to the competitive relationship between SFA and MSA. Thirdly, the contribution of the SFA-involved cluster formation pathway was positively associated with the concentration of [SFA] (Fig. 8 (d))"

Rather than fixing the concentrations of other precursors and discussing the impact of changes in a single component's concentration, I think it would be more valuable to explore the specific nucleation mechanisms in regions such as India or China by incorporating observational concentrations of SFA, MSA, and MA as reported in field studies.

**Comment 5**: The boundary of the ACDC simulation is the smallest clusters that can be stable enough to

grow outside of the simulated system. What's the boundary of the present ACDC simulation?

**Minor Comments:**

**Comment 1 Page 3 line 89**: "Due to the concentration of SA …. , MSA-driven NPF has attracted growing attention"

Please use either "MSA" or "CH₃SO₃H" consistently to represent methanesulfonic acid. The same issue also appears on representation of sulfamic acid.

**Comment 2 Page 4 line 107-108**: "Atmospheric Clusters Dynamic Code (ACDC) models to evaluate the potential effect of SFA on nucleation and NPF."

Please cite the original publications of ACDC models. Additionally, cite some research to demonstrate the reliability of this method.

**Comment 3 Page 17 line 473-478**:

Some references include article links, while others do not. Please unify the reference format.

**Comment 4 Page 28 line 691-692**:

The y-axis in Figure 6 contains too much information. It is recommended to adjust the layout to make the results more visually concise.

---

## Author Response (AR2)

**Responses to Referee #1's comments**

We are grateful to the reviewers for their valuable and helpful comments on our manuscript "A novel formation mechanism of sulfamic acid and its enhancing effect on methanesulfonic acid-methylamine aerosol particle formation in agriculture-developed and coastal industrial areas" (Manuscript ID: EGUSPHERE-2024-2638). We have revised the manuscript carefully according to reviewers' comments. The point-to-point responses to the Referee #1's comments are summarized below:

**Referee Comments**

The manuscript egusphere-2024-2638, "A novel formation mechanism of $NH_2SO_3H$ and its enhancing effect on methanesulfonic acid-methylamine aerosol particle formation in agriculture-developed and coastal industrial areas". The work studied the formation of sulfamic acid via $HNSO_2$ hydrolysis in the gas phase and at the air-water interface by using theoretical methods. Then, the author investigated the new particle formation for the role of sulfamic acid in $CH_3SO_3H$-$CH_3NH_2$ system. The work is very interesting for understanding the chemical processes of sulfamic acid in the atmosphere. However, there are some issues that should be addressed before publication.

**Response:** We would like to thank the reviewer for the positive and valuable comments, and we have revised our manuscript accordingly.

**Major issues**

**Comment 1.**

In Line 39, "the concentration of $NH_2SO_3H$ was expected to reach up to$10^8$molecules·cm$^{-3}$", the concentration of sulfamic acid was only estimated by theoretical method, not measured by field observations. Therefore, it is required to elucidate this point.

**Response:** Thanks for the suggestion of the reviewer. As the suggestion of the reviewer, in Lines 38-40 Page 2 of the revised manuscript, the sentence of "the atmospheric concentration of $NH_2SO_3H$ was expected to reach up to$10^8$ molecules·cm$^{-3}$ (Li et al., 2018)." has been changed as "the atmospheric concentration of SFA estimated by theoretical method of CCSD(T)-F12/cc-pVDZ-F12//M06-2X/6-311++G(3$df$,3$pd$) (Li et al., 2018) was expected to reach up to$10^8$ molecules·cm$^{-3}$".

**Comment 2.**

Lines 41-42, "the sources of $NH_2SO_3H$ in the atmosphere have been well investigated (Lovejoy and Hanson, 1996; Pszona et al., 2015; Li et al., 2018; Larson and Tao, 2001; Manonmani et al., 2020; Zhang et al., 2022)." In fact, sulfamic acid has been not investigated by field measurements. Therefore, it is not well investigated in the atmosphere.

**Response:** Thanks for the suggestion of the reviewer. Indeed, it is true that SFA has not been measured in the field. Therefore, atmospheric sulfamic acid has not been well studied. In Lines 41-43 Page 2 of the revised manuscript, "So, the sources of $NH_2SO_3H$ in the atmosphere have been well investigated (Lovejoy and Hanson, 1996; Pszona et al., 2015; Li et al., 2018; Larson and Tao, 2001; Manonmani et al., 2020; Zhang et al., 2022)." has been changed as "So, the sources of SFA in the atmosphere has been focused by several groups (Lovejoy and Hanson, 1996; Pszona et al., 2015; Li et al., 2018; Larson and Tao, 2001; Manonmani et al., 2020; Zhang et al., 2022).".

**Comment 3.**

Lines 46-47, "for the hydrolysis of $SO_3$ assisted by water molecule ($10^{-11}$-$10^{-10}$ cm$^3$ molecule$^{-1}$ s$^{-1}$) (Kim et al., 1998; Hirota et al., 1996; Shi et al., 1994)." Some important references are missing such as J. Am. Chem. Soc. 2023, 145, 19866-19876. and J. Am. Chem. Soc. 1994, 116, 10314−10315.

**Response:** Thanks for the suggestion of the reviewer. We apologize for missing some important references. As the suggestion of the reviewer, some important references have been added in Lines 46-49 Page 2 of the revised manuscript, which has been organized as "which was close to the value for the hydrolysis of $SO_3$ assisted by water molecule ($10^{-11}$-$10^{-10}$ cm$^3$ molecule$^{-1}$ s$^{-1}$) (Kim et al., 1998; Hirota et al., 1996; Shi et al., 1994; Kolb et al., 1994; Long et al., 2013; Long et al., 2023; Ding et al., 2023; Cheng et al., 2023; Wang et al., 2024).".

**Comment 4.**

What is the concertation of $HNSO_2$ in the atmosphere? This is very necessary for determining the importance of $HNSO_2$ in the atmosphere.

**Response:** Thanks for the suggestion of the reviewer. As the suggestion of the reviewer, we have conducted an extensive review of the relevant literature. However, the concentrations of $HNSO_2$ in the atmosphere have not been reported. As the absence of the concentration of $HNSO_2$, the competitiveness between MSA-assisted $HNSO_2$ hydrolysis and the $NH_3$-assisted ammonolysis of $SO_3$ (the traditional source of SFA) cannot be further confirmed. The related discussion has been found in Line 236 Page 8 to Line 237 Page 9 of the revised manuscript, which has been organized as "However, due to the absence of the concentration of $HNSO_2$, the competitiveness of these two reactions cannot be further confirmed." Although the concentration of $HNSO_2$ has not been reported, it is still important to study $HNSO_2$ hydrolysis with MSA in the gas phase and at the air-water interface. The detailed importance of $HNSO_2$ hydrolysis with MSA has been presented as follows.

In the gas phase, with the significant decrease in atmospheric water molecules with increasing altitude, MSA has a significantly greater catalytic ability than $H_2O$ in accelerating the rate of $HNSO_2$ hydrolysis within 5-15 km. At the air-water interface, two types of reactions, the ions forming mechanism and the proton exchange mechanism to form $NH_2SO_3^-\cdots H_3O^+$ ion pair were observed on the timescale of picosecond, which is at least two orders of magnitude faster than the corresponding gas-phase reaction. Nobly, considering the overall environment of sulfuric acid emission reduction, the present findings suggest that SFA may play a significant role in NPF and the growth of aerosol particles as $i$) SFA can directly participate in the formation of MSA-$CH_3NH_2$-based cluster and enhance the rate of NPF from these clusters by approximately $10^3$ times at 278.15 K; and $ii$) the $NH_2SO_3^-$ species at the air-water interface can attract gaseous molecules to the aqueous surface, and thus promote particle growth.

**Comment 5.**

The reliability of the chosen methods should be clarified in the $HNSO_2$ + $CH_3SO_3H$ reaction. Although the traditional method CCSD(T)//M06-2X has been widely used for atmospheric reactions, it should be noted that there are quite large uncertainties for estimating barrier height. This should clearly tell the potential readers.

**Response:** Thanks for the suggestion of the reviewer. As the suggestion of the reviewer, the reliability of CCSD(T)/aug-cc-pVDZ//M06-2X/6-311+G(2$df$,2$pd$)-based calculation method has been verified as follows. Firstly, the geometry and frequency calculation involved in the $HNSO_2$ hydrolysis were verified (Fig. S2) at three different theoretical levels of M06-2X/6-311++G(3$df$,2$pd$), M06-2X/6-311++G(3$df$,3$pd$) and M06-2X/aug-cc-pVTZ and experimental values. Then, based on the M06-2X/6-311++G(2$df$,2$pd$) optimized geometries, the corresponding single point energy calculations (Table S1) were performed at the CCSD(T)-F12/cc-pVDZ-F12,

CCSD(T)-F12/cc-pVTZ-F12, CCSD(T)/CBS and CCSD(T)/aug-cc-pVTZ levels, respectively. The main revision has been made as follows.

[Figure]

**Fig. S2** The optimized geometrical structures for the species involved in the HNSO$_2$ hydrolysis at several different levels of theory.

[a, b and, c] respectively represents the values obtained at the M06-2X/6-311++G(3*df*,2*pd*), M062X/6-311++G(3*df*,3*pd*) and M06-2X/aug-cc-pVTZ level of theory, [d] represents the experimental values (The values in parentheses were obtained at the M06-2X/6-311++G(2*df*,2*pd*) level of theory; bond length is in angstrom and angle is in degree.).

(a) The geometric parameters of the reactants of HNSO$_2$, H$_2$O and NH$_2$SO$_3$H (SFA) have been displayed in Fig. S2. As seen in Fig. S2, the mean absolute deviation of calculated bond distances and bond angles between the M06-2X/6-311++G(2*df*,2*pd*) level and the experimental reports were

0.02 Å and 0.57°, respectively. This reveals that the calculated bond distances and bond angles at the M06-2X/6-311++G(2*df*,2*pd*) level agree well with the available experimental values (From the pubchem database, https://pubchem.ncbi.nlm.nih.gov/#opennewwindow.). In addition, we have reoptimized all equilibrium structures of HNSO$_2$, H$_2$O and NH$_2$SO$_3$H at three different theoretical levels of M06-2X/6-311++G(3*df*,2*pd*), M062X/6-311++G(3*df*,3*pd*) and M06-2X/aug-cc-pVTZ

levels. For the calculated geometrical parameters of these species, the mean absolute deviation of calculated bond distances and bond angles between the M06-2X/6-311+G(2*df*,2*pd*) level and the other levels were within 0.02 Å and 0.2°, respectively. Therefore, due to its efficiency, the M06-

2X/6-311++G(2*df*,2*pd*) was adopted to optimize the geometries of all stationary points involved in the HNSO$_2$ hydrolysis. Based on this, in Lines 121 to 124 Page 5 of the revised manuscript, the sentence of "It is noted that the calculated bond distances and bond angles at the M06-2X/6-

311++G(2*df*,2*pd*) level (Fig. S2) agree well with the available values (Fig. S2) from the experiment and three different theoretical levels of M06-2X/6-311++G(3*df*,2*pd*), M062X/6-311++G(3*df*,3*pd*)

and M06-2X/aug-cc-pVTZ levels." has been added.

**Table S1** The Energy barriers ($\Delta E$) and unsigned error (UE) (kcal·mol$^{-1}$) for the HNSO$_2$ hydrolysis at different theoretical the potential energy profile ($\Delta G$) correction

| Methods | $\Delta E$ [a] | $\Delta E$ [b] | $\Delta E$ [c] | UE |
|---------|------|------|------|------|
| CCSD(T)/CBS//M06-2X/ 6-311++G(2$df$,2$pd$) | 3.4 | 29.7 | -23.0 | 0.00 |
| CCSD(T)-F12/cc-pVDZ-F12//M06-2X/ 6-311++G(2$df$,2$pd$) | 3.6 | 30.6 | -22.0 | 0.71 |

[a, b and c] respectively denote the species of pre-reactive complexes, transition states and products involved in the HNSO$_2$ hydrolysis.

(b) To further confirm the reliability of the CCSD(T)-F12/cc-pVDZ-F12//M06-2X/6-311++G(2$df$,2$pd$) level of theory, single-point energy calculations for the HNSO$_2$ hydrolysis in the gas phase have been performed at two different levels of CCSD(T)/CBS and CCSD(T)-F12/cc-pVDZ-F12 based on the optimized geometries at the M06-2X/6-311++G(2$df$,2$pd$) level. Notably, the complete basis set (CBS) obtained by basis set extrapolation is used as the reference basis set. As presented in Table S1, compared with unsigned error calculated at the CCSD(T)/CBS//M06-2X/6-311++G(2$df$,2$pd$) level, unsigned errors calculated at CCSD(T)-F12/cc-pVDZ-F12//M06-2X/6-311++G(2$df$,2$pd$) was 0.71 kcal·mol$^{-1}$. This suggests that the relative energies obtained at the CCSD(T)/aug-cc-pVDZ//M06-2X/6-311+G(2$df$,2$pd$) level was reasonable. Considering the computational accuracy and cost, the CCSD(T)/aug-cc-pVDZ//M06-2X/6-311+G(2$df$,2$pd$) method was chosen to calculate the single point energies of all the species involved in the HNSO$_2$ hydrolysis. Thus, in Lines 130 to 133 Page 5 of the revised manuscript, the sentence of "The CCSD(T)/aug-cc-pVDZ method was chosen to calculate the relative energies as the fact that, compared with unsigned error (Table S1) calculated at the CCSD(T)/CBS//M06-2X/6-311++G(2$df$,2$pd$) level, unsigned errors calculated at CCSD(T)-F12/cc-pVDZ-F12//M06-2X/6-311++G(2$df$,2$pd$) was 0.71 kcal·mol$^{-1}$." has been added.

**Comment 6.**

In kinetics calculations, it is unclear. There are lots of issues that must be addressed. Provide the details of VRC calculations. For example, how to set pivot points and what is the electronic structure method for VRC-TST calculations? The author should provide the input files for VRC-TST and MESMER calculations in Supporting information to help the potential readers to understand the computational details.

**Response:** Thanks for the suggestion of the reviewer. The pivot point setting method and the electronic structure method for VRC-TST calculation are provided in detail (shown in Part S1 in the

Supplement). Meanwhile, the input files for VRC-TST and MESMER calculations have been provided in Supporting information. The main revision has been made as follows.

(a) Herein, we describe the implementation details of the VRC-TST calculation in in Part S1

in the Supplement. Specifically, there are two assumptions in VRC-VTST calculation: (1) the contribution of the vibrational modes of reactants to the partition function is canceled by the corresponding contribution of transition states to the partition function; (2) the internal geometries of reactants are fixed along the reaction coordinate. The reaction coordinate in VRC-VTST is different from that in RP-VTST and determined by the pivot points of each reactant fragment. For the $HNSO_2$ hydrolysis reaction, the pivot points of $HNSO_2$ (points 1 and 2) are located at a distance

±d along its S axis. Meanwhile, the pivots of $H_2O$ (points 3 and 4) are located at a distance ±d perpendicular to $H_2O$ molecule lane. As shown in Fig. S6, the Multiwfn package combined with the

VMD software is adopted to visualize the reaction system and help determine the location of pivot points. The reaction coordinate value ($s$) is defined as the minimum of the distance ($r_{ij}$) between the pivot point $i$ (=1 or 2) and pivot point $j$ (=3 or 4), where $i$ and $j$ represent the pivot points of $HNSO_2$

and $H_2O$ molecules, respectively. Hence, each of the four dividing surfaces is obtained by symmetrically placing two pivot points of each radical fragment (1-3, 1-4, 2-3, and 2-4). For example, if the reaction coordinate $s$ is equal to $r_{23}$, one of the four dividing surfaces (2-3), is determined by the locations of pivot points 2, 3 and the reaction coordinate $s$. There are total four pair of pivot points, the other three dividing surfaces (1-3, 1-4, 2-4) are defined by their corresponding pivot points and reaction coordinates $s$. Note that the locations of pivot points are critical to the rate constant calculation. Considering the difference between $HNSO_2$ and $H_2O$

molecules, the distance $s$ between pivot points varies from 2.5 to 6 Å for $HNSO_2$ and $H_2O$ in each case with a 0.5 Å grid increment. So, in Lines 139 to141 Page 5 of the revised manuscript, the sentence of the "Meanwhile, two pivot points (Bao et al., 2016; Long et al., 2021; Georgievskii and

Klippenstein, 2003; Meana-Pañeda et al., 2024) were selected to calculate the high-pressure limiting rate for the $HNSO_2$ hydrolysis (shown in Part S1 in the Supplement)." has been added. Also, the computational details of VRC-VTST calculations have been added in Line 159 Page S15to 180 Page

S16 of the revised Supplement.

[Figure]

     **Fig. S6** The placements of the pivot points for the $HNSO_2$ hydrolysis (b) The electronic structure method for VRC-TST calculations is based on Gaussian 09

program using the M06-2X/6-311++G(2$df$,2$pd$). So, in Lines 137-139 Page 5 of the revised manuscript, the sentence of the "It's worth noting that the electronic structure method for VRC-TST

calculations is based on Gaussian 09 program using the M06-2X/6-311++G(2$df$,2$pd$)." has been added.

(c) The input files for VRC-TST and MESMER calculations have been provided in Supplement.

**Comment 7.**

According to the authors' previous research (Phys. Chem. Chem. Phys., 2022, 24, 4966-4977), the reaction of $HNSO_2$ with $nH_2O$ also has a sufficiently low free energy barrier, which implies that

$HNSO_2$ can undergo hydrolysis or decomposition directly at the gas-liquid interface or in the bulk phase. This seems to contradict the explanation on line 228 (page 8), given that the concentration of water is sufficiently high.

**Response:** Thanks for the suggestion of the reviewer. According to the previous work (***Phys. Chem.***

***Chem. Phys.***, 2022, 24, 4966-4977), the hydrolysis of $HNSO_2$ assisted by $H_2O$, $(H_2O)_2$ and $(H_2O)_3$

involved a loop structure mechanism. These reactions were known to occur via the initial formation of ring hydrogen bonding complex $HNSO_2\cdots(H_2O)_n$ ($n$ = 1-3) with the calculated relative free energy of 0.2-3.6 kcal·mol$^{-1}$ followed by their rearrangement to form $NH_2SO_3H$. As the higher entropy effect, hydrogen bonding complex $HNSO_2\cdots(H_2O)_n$ ($n$ = 1-3) were formed hardly under actual atmospheric conditions, and thus the loop structure mechanism for the hydrolysis of $HNSO_2$

assisted by $H_2O$, $(H_2O)_2$ and $(H_2O)_3$ is not easy to occur in the gas phase. This is similar with

$CH_3SO_3H$-assisted gaseous hydrolysis of $HNSO_2$ which does not occur within the 100 ps.

At the air-water interface, the $HNSO_2$ molecule is stable and does not dissociate within 10 ps, where the loop structure of hydrogen bonding complex $HNSO_2\cdots(H_2O)_n$ ($n = 1$-3) has not been observed. This is proved by the BOMD simulation illustrated in Fig. S8 where the hydrated form of

$HNSO_2$ was not conducive to $HNSO_2$ hydrolysis at the air-water interface. So, even if the concentration of water molecules at the air-water interface is sufficiently high, the probability that

$HNSO_2$ can be hydrolyzed or decomposed either directly at the air-water interface or in the bulk phase is small. This is agreed with the simulation results. Based on this analysis above, in Lines

272-274 Page 10 of the revised manuscript, the sentence of the "Meanwhile, although $HNSO_2$

remains stable at the air-water interface (seen in Fig. S8) and does not dissociate within 10 ps, the hydrated form of $HNSO_2$ illustrated in Fig. S8 was not conducive to $HNSO_2$ hydrolysis at the air- water interface." has been added to prove that the hydrated form of $HNSO_2$ was not conducive to

$HNSO_2$ hydrolysis at the air-water interface.

**Comment 8.**

Why did the authors not consider a third access channel in the gas phase, that is, the reaction pathway of $HNSO_2\cdots CH_3SO_3H + H_2O$? Considering the reactions at the gas-liquid interface, it seems more plausible that $HNSO_2\cdots CH_3SO_3H$ would first form a complex before reacting with water molecules.

Considering the reactions at the gas-liquid interface, it seems more plausible that $HNSO_2\cdots$

$CH_3SO_3H$ would first form a complex before reacting with water molecules.

**Response:** Thanks for the suggestion of the reviewer. Indeed, the reaction pathway of $HNSO_2\cdots$

$MSA + H_2O$ is feasible. However, the concentration of water molecules in the atmosphere is about

$10^{18}$ molecules·$cm^{-3}$, which is much higher than those of $HNSO_2$ and MSA ($10^5$-$10^9$ molecules·cm$^-$

$^3$). Considering the harsh conditions for the initial formation of dimers between $HNSO_2$ and MSA

(i.e., $HNSO_2$ and MSA are sufficiently concentrated in the atmosphere.), we predict that the primary preliminary dimers will continue to be dominated by $HNSO_2\cdots H_2O$ and $MSA\cdots H_2O$ complexes.

So, in Line 203 Page 7 to Line 207 Page 8 of the revised manuscript, the sentence of the "As the concentration of water molecule ($10^{18}$ molecules·$cm^{-3}$ (Anglada et al., 2013)) in the atmosphere is much higher than those of $HNSO_2$ and MSA ($10^5$-$10^9$ molecules·cm$^{-3}$ (Shen et al., 2020)), the reaction pathway of $HNSO_2\cdots MSA + H_2O$ is hard to occur in actual atmospheric conditions. So,

Channel MSA proceeds through the initial formation of dimers ($HNSO_2\cdots H_2O$ and $MSA\cdots H_2O$)

via collisions between $HNSO_2$ (or MSA) and $H_2O$." has been added.

**Comment 9.**

In Section 3.3, the authors examined the impact of MSA-MA-SFA clusters on nucleation.

Interestingly, DMA, which has a stronger nucleation capability, and $NH_3$, which has a higher concentration, were excluded. I would like the authors to provide some appropriate justifications for this.

**Response:** Thanks for the suggestion of the reviewer. Previous studies have demonstrated that

MSA-driven new particle formation (NPF) has attracted growing attention, as MSA significantly contributes to NPF in scenarios with only natural sources of $SO_2$ were present. Currently, atmospheric bases, including methylamine (MA), monoethanolamide, and dimethylamine (DMA), have a key role in MSA-driven aerosol particle generation and growth, where MA exhibits the strongest enhancing capability (***Environ. Sci. Technol***, 2017, 51, 243-252; ***J. Phys. Chem. B***, 2016,

120, 1526-1536; ***Environ. Sci. Technol.***, 53, 14387-14397, 2019; ***Atmos. Environ.,*** 2023, 311,

120001). So, we choose MA over DMA and $NH_3$. This choice is similar to that previously reported in the relevant references (***Atmos. Chem. Phys.,*** 22, 2639-2650; 2022***Atmos. Environ.,*** 2023, 311,

120001). Based on this analysis above, in Lines 94-96 Page 4 of the revised manuscript, the sentence of the "Initially, the binary nucleation of MSA with inorganic ammonia and organic amines in the atmosphere has been reported, where MA exhibits the strongest enhancing capability (Chen et al.,

2016; Chen and Finlayson-Pitts, 2017; Shen et al., 2019; Hu et al., 2023)." has been reorganized to prove that the MSA-MA system was chosen over MSA-DMA.

**Comment 10.**

Since the ammonolysis of $SO_3$ is the primary pathway for SFA formation, the authors could have compared it with the current pathway, which would be necessary for accurately assessing the atmospheric significance of the current reaction.

**Response:** Thanks for the suggestion of the reviewer. As the suggestion of reviewer, we compared the $NH_3$-assisted ammonolysis of $SO_3$ with the MSA-assisted $HNSO_2$ hydrolysis. In Line 233 Page

8 to Line 237 Page 9 of the revised manuscript, "Besides, MSA-assisted $HNSO_2$ hydrolysis is reduced by 4.9 kcal·mol$^{-1}$ in energy barrier than the $NH_3$-assisted ammonolysis of $SO_3$ with its rate constant at 298 K ($2.85 \times 10^{-11}$ cm$^3$·molecule$^{-1}$·s$^{-1}$) close to the value of ammonolysis of $SO_3$ with

$NH_3$ ($4.35 \times 10^{-10}$ cm$^3$·molecule$^{-1}$·s$^{-1}$) (Li et al., 2018). However, due to the absence of the concentration of $HNSO_2$, the competitiveness of these two reactions cannot be further confirmed."

has been added.

**Comment 11.**

In Fig. 6b and Fig. 7b, it is necessary for the authors to carefully examine whether the significant abrupt changes caused by the concentrations of SFA and MA are reasonable.

**Response:** Thanks for the suggestion of the reviewer. It is noted that in Fig. 6(b), due to the competitive relationship between MSA and SFA, at low concentrations of SFA, the binding capacity of MSA with MA is stronger than that of SFA with MA, resulting in only a small amount of SFA

participating in cluster formation. However, as the concentration of SFA increases, the number of

$(MSA)_x·(MA)_y·(SFA)_z$ (where $y \leq x + z \leq 3$) ternary clusters increases, leading to the formation of more hydrogen bonds and a significant increase in $R_{SFA}$. Similarly, in Fig. 7(b), at a certain concentration of SFA and MA, as the concentration of MSA increases, the hydrogen bonds between

SFA and MA are disrupted, leading to more binding of MA and MSA rather than SFA, resulting in a sharp decrease in $R_{SFA}$. In Lines 364-369 Page 12 of the revised manuscript, "It is noted that in

Fig. 6(b), due to the competitive relationship between MSA and SFA, at low concentrations of SFA, the binding capacity of MSA with MA is stronger than that of SFA with MA, resulting in only a small amount of SFA participating in cluster formation. However, as the concentration of SFA

increases, the number of $(MSA)_x·(MA)_y·(SFA)_z$ (where $y \leq x + z \leq 3$) ternary clusters increase, leading to the formation of more hydrogen bonds and a significant increase in $R_{SFA}$." has been added.

**Comment 12.**

In the introduction, the authors mention that the $pK_a$ may affect the transfer of protons, thereby affecting the catalytic ability. Whether similar trends will also directly affect the nucleation capability should be considered, such as in the cases of MSA-MA-SFA, MSA-MA-SA, and SA/FA-

MA-SFA.

**Response:** Thanks for the suggestion of the reviewer. We apologize for the misunderstanding about

$pK_a$ in Lines 63-65 Pages 2-3. Indeed, our aim is to illustrate the importance of MSA as a catalyst from $pK_a$ perspective. In order not to create ambiguity, as for the discussions of $pK_a$, the sentence of the "It was noted that as the acidity of $CH_3SO_3H$ ($pK_a$ = -1.92) was significantly stronger than that of water ($pK_a$ = 15.0) and formic acid ($pK_a$ = 3.74), it may be predicted that the proton transfer reaction for the hydrolysis of $HNSO_2$ with $CH_3SO_3H$ was much easier than those with water and formic acid. It was also noted that although $CH_3SO_3H$ was less acidic than $H_2SO_4$ ($pK_a$ = -3.00), with the global reduction in the concentration of $H_2SO_4$ resulting from $SO_2$ emission restrictions, the contribution of $CH_3SO_3H$ to aerosol nucleation has received the widespread attention of scientists." had been deleted. Meanwhile, the importance of MSA as a catalyst in $HNSO_2$ hydrolysis has been organized as "It was noted that, with the global reduction in the concentration of $H_2SO_4$

resulting from $SO_2$ emission restrictions, the contribution of MSA to aerosol nucleation has received the widespread attention of scientists." in Lines 63-66 Page 3 of the revised manuscript.

 **Responses to Referee #2's comments**

 We are grateful to the reviewers for their valuable and helpful comments on our manuscript "A

 novel formation mechanism of sulfamic acid and its enhancing effect on methanesulfonic acid-

 methylamine aerosol particle formation in agriculture-developed and coastal industrial areas"

 (Manuscript ID: EGUSPHERE-2024-2638). We have revised the manuscript carefully according to

 reviewers' comments. The point-to-point responses to the Referee #2's comments are summarized

 below:

 **Referee Comments**

 Wang et al. present a novel formation mechanism of sulfamic acid ($NH_2SO_3H$) and its

 enhancement effect in methanesulfonic acid-methylamine (MSA-MA) aerosol particle formation.

 The study centers on the production, consumption, and potential pollution impacts of sulfamic acid

 over agriculture-intensive and coastal industrial regions. The most part of this manuscript is well

 written and of broad interest to the readership of *Atmospheric Chemistry and Physics*. I recommend

 publication in *Atmospheric Chemistry and Physics* after the following comments have been

 addressed.

 **Response:** We would like to thank the reviewer for the positive and valuable comments, and we

 have revised our manuscript accordingly.

 **Major issues**

 **Comment 1.**

 Pages 2- 3 lines 57-62: "As the direct hydrolysis of $HNSO_2$ with a high energy barrier takes place

 hardly in the gas phase, the addition of a second water molecule, formic acid and sulfuric acid

 ($H_2SO_4$, SA) have been proved to promote the product of $NH_2SO_3H$ through the hydrolysis of

 $HNSO_2$. However, to the best of our knowledge, the gaseous hydrolysis of $HNSO_2$ with $CH_3SO_3H$

 has not yet been investigated"

 The necessity for studying the gaseous hydrolysis of $HNSO_2$ with $CH_3SO_3H$ is not sufficiently

 clarified. Is there any research or evidence indicating that the reaction processes you introduced

 earlier are insufficient to explain the source of sulfamic acid? If so, please provide additional

 information.

**Response:** Thanks for the suggestion of the reviewer. We apologize for not explicitly studying the necessity for studying the gaseous hydrolysis of $HNSO_2$ with MSA. According to the reviewer's suggestion, the main revision of the necessity for studying the gaseous hydrolysis of $HNSO_2$ with MSA has been made as follows.

(a) In fact, the gaseous hydrolysis of $HNSO_2$ with MSA was very important at two points. Firstly, with the global reduction in the concentration of $H_2SO_4$ resulting from $SO_2$ emission restrictions, the contribution of MSA to aerosol nucleation has received the widespread attention of scientists. As a major inorganic acidic air pollutant (***Chemosphere.***, 2020, 244, 125538-125547), the concentration of MSA in the atmosphere was noted to be notably high across various regions, spanning from coastal to continental, with levels found to be between 10% and 250% of those measured for SA (***Environ. Sci. Technol.***, 2019 53, 14387-14397; ***Environ. Sci. Technol.***, 2020, 54, 13498-13508; ***J. Phys. Chem. A***, 2014 118, 5316-5322; ***Atmos. Environ.***, 2023, 311, 120001). Based on the analysis above, the importance of MSA has been reorganized as "It was noted that, with the global reduction in the concentration of $H_2SO_4$ resulting from $SO_2$ emission restrictions, the contribution of MSA to aerosol nucleation has received the widespread attention of scientists. As a major inorganic acidic air pollutant (Chen et al., 2020), the concentration of MSA in the atmosphere was noted to be notably high across various regions, spanning from coastal to continental, with levels found to be between 10% and 250% of those measured for SA (Shen et al., 2019; Dawson et al., 2012; Bork et al., 2014; Shen et al., 2020; Berresheim et al., 2002; Hu et al., 2023)." in the Lines 63-70 Page 3 of the revised manuscript. Secondly, the gaseous hydrolysis of $HNSO_2$ with MSA has not yet been investigated, which will confine the understanding for the source of SFA in regions with significant pollution and high levels of MSA. So, the necessity for studying the gaseous hydrolysis of $HNSO_2$ with MSA has been added as "However, to the best of our knowledge, the gaseous hydrolysis of $HNSO_2$ with MSA has not yet been investigated, which will confine the understanding for the source of SFA in regions with significant pollution and high levels of MSA." in Lines 70-72 Page 3 of the revised manuscript.

(b) The traditional view is that the source of sulfamic acid primarily originates from the ammonolysis of $SO_3$ in the troposphere, which has been widely reported by many groups (***J. Am. Chem. Soc***, 2018,140, 11020-11028; ***J. Phys. Chem. A***, 2019,123 14, 3131-3141; ***J. Mass Spectrom.***, 50, 127-135, 2015). In addition to the traditional source of sulfamic acid, the hydrolysis of $HNSO_2$ has garnered increasing attention as a potential new source of sulfamic acid.

Consequently, the hydrolysis of $HNSO_2$ with MSA has been studied in this paper. To date, the atmospheric concentration of sulfamic acid has only been estimated in the $SO_3$-$NH_3$ system by Li et al (***J. Am. Chem. Soc,*** 2018,140, 11020-11028) using the theoretical method, no field observations of atmospheric sulfamic acid concentrations have been reported. So, the contribution of the $HNSO_2$ hydrolysis with MSA to atmospheric sulfamic acid sources remains uncertain.

However, "A novel formation mechanism of sulfamic acid and its enhancing effect on methanesulfonic acid-methylamine aerosol particle formation in agriculture-developed and coastal industrial areas" not only elucidates a novel mechanism underlying the hydrolysis of $HNSO_2$ with

MSA, but also highlight the potential contribution of sulfamic acid on aerosol particle growth and new particle formation.

**Comment 2.**

Page 6 lines 155-156: "The ACDC model was utilized to simulate the $(SFA)_x(MSA)_y(MA)_z$ ($0 \leqslant$

$z \leqslant x + y \leqslant 3$) cluster formation rates and explore the potential mechanisms". The structural stability of clusters directly impacts the nucleation ability of a multi-components system. How was the most stable structure of $(SFA)_x(MSA)_y(MA)_z$ ($0 \leqslant z \leqslant x + y \leqslant 3$) clusters used in this paper obtained?

**Response:** Thanks for the suggestion of the reviewer. The most stable structure of

$(SFA)_x(MSA)_y(MA)_z$ ($0 \leqslant z \leqslant x + y \leqslant 3$) clusters were searched with ABCluster software (Zhang and Dolg, 2015). In Lines 169-173 Page 6 of the revised manuscript, the sentence of "The

ACDC model was utilized to simulate the $(SFA)_x(MSA)_y(MA)_z$ ($0 \leqslant z \leqslant x + y \leqslant 3$) cluster formation rates and explore the potential mechanisms." has been reorganized as "The ACDC model (McGrath et al., 2012; Hu et al., 2023; Zhao et al., 2020; Zhang et al., 2024; Tsona Tchinda et al., 2022; Liu et al., 2020) was utilized to simulate the $(MSA)_x(MA)_y(SFA)_z$ ($0 \leq y \leq x + z \leq 3$)

cluster formation rates and explore the potential mechanisms, where the most stable structure of

$(SFA)_x(MSA)_y(MA)_z$ ($0 \leqslant z \leqslant x + y \leqslant 3$) clusters were searched with ABCluster software (Zhang and Dolg, 2015) (The details in Part S2 of the Supplement).". In Part S2 of the Supplement, the specific steps of configurational sampling have been added as "A multistep global minimum sampling scheme, which has previously been applied to study the atmospheric cluster formation, was employed to search for the global minima of the $(SFA)_x(MSA)_y(MA)_z$ ($0 \leqslant z \leqslant x + y$

$\leqslant 3$) clusters. To locate the global minimum energy structure, the artificial bee colony algorithm was systematically employed by the ABCluster program to generate $n \times 1000 (1 < n$

$\leq 4$) initial random configurations for each cluster, and then, PM6 semi-empirical method was used to further pre-optimize the produced configurations above. Second, up to 100 structures with relatively lower energies were selected from the $n \times 1000$ structures (where $1 < n \leq 4$), and a M06-2X/6-31+G($d,p$) level of theory was applied for subsequent optimization. Finally, further geometry optimization and frequency calculations at the M06-2X/6-311++G($2df,2pd$)

level of theory were performed to optimize the 10 best of 100 optimized configurations, and then the global minimum structure with the lowest energy was obtained. Subsequently, the

M06-2X function combined with the 6-311++G($2df,2pd$) basis set was chosen as it has been proven to be accurate in estimating the thermodynamic properties of atmospheric clusters, such as organic acid-SA-amine clusters, amide-SA clusters or amino acid-SA clusters. In this study, all the density functional theory (DFT) calculations were implemented in the Gaussian 09

program.".

**Comment 3.**

Thermodynamic parameters, obtained from quantum chemical calculations executed at the M06-

2X/6-311++G(*2df,2pd*) level, were used as inputs for the ACDC model. Please further justify for why the M06-2X/6-311++G(*2df,2pd*) level of theory was employed to obtain the thermodynamic parameters used as inputs for the ACDC model.

**Response:** Thanks for your valuable comments. Many benchmark studies (***Atmos. Chem. Phys.,***

2024, 24, 3593-3612; ***Atmos. Chem. Phys.,*** 2021, 21, 6221-6230; ***Atmos. Chem. Phys.,*** 2022, 22,

1951-1963; ***Sci. Total Environ.,*** 2020, 723, 137987) show that the M06-2X functional has good performance compared to other common functionals for gaining the Gibbs free energies. For all the

M06-2X calculations with the 6-311++G($2df,2pd$) basis set was used, as it is a good compromise between accuracy and efficiency and does not yield significant errors in the thermal contribution to the free energy compared to much larger basis sets such as 6-311++G($3df,2pd$). So, according to the reviewer's suggestion, the sentence of "Notably, many benchmark studies (Zhao et al., 2020; Zhang et al., 2024; Tsona Tchinda et al., 2022; Liu et al., 2020) show that the M06-2X functional has good performance compared to other common functionals for gaining the Gibbs free energies. For all the

M06-2X calculations with the 6-311++G($2df$,$2pd$) basis set was used, as it is a good compromise between accuracy and efficiency and does not yield significant errors in the thermal contribution to the free energy compared to much larger basis sets such as 6-311++G($3df$,$3pd$), with the differences of relative $\Delta G$ less than 1.75 kcal·mol$^{-1}$ (Table S7)." was added in Line 177 Page 6 to line 183 Page

7 of the revised manuscript. Besides, for the optimized geometries of the important precursors of atmospheric aerosol nucleation (MSA, MA and SFA), the main bond lengths and bond angles at two different theoretical levels of M06-2X/6-311++G($2df$,$2pd$) and M06-2X/6-311++G($3df$,$3pd$)

has been listed in Fig. S17. Moreover, in Table S7, the predicted relative $\Delta G$ of MSA·MA,

SFA·MA, MSA·SFA and MSA·SFA·MA clusters at the M06-2X/6-311++G($2df$,$2pd$) level was compared with the corresponding values at the M06-2X/6-311++G($3df$,$3pd$) level. Based on the above analysis, the corresponding changes are as follows.

(a) For the MSA·A, SFA·A, MSA·SFA and MSA·SFA·MA clusters, the geometric parameters (Fig. S15) at the M06-2X/6-311++G($3df$,$3pd$) and M06-2X/6-311++G($2df$,$2pd$) levels of theory were calculated. The geometrical structure analysis indicated that the bond lengths and angles obtained from both theoretical levels are close to each other. So, all optimizations and vibrational frequency were calculated at M06-2X/6-311++G($2df$,$2pd$) level.

[Figure]

**Fig. S17** The optimized geometries of the important precursors of atmospheric aerosol nucleation (MSA, MA and SFA), especially the main bond lengths and bond angles at two different theoretical levels. SFA, MSA and MA are the shorthand for formic acid, sulfuric acid and ammonia, respectively. [a] The values obtained at the M06-2X/6-311++G(2*df*,2*pd*) level of theory. [b] The values obtained at the M06-2X/6-311++G(3*df*,3*pd*) level of theory. Bond length is in angstrom and angle is in degree (b) We calculated the Gibbs free energy (in Table S7) for the MSA·MA, SFA·MA, MSA·SFA and MSA·SFA·MA clusters at the M06-2X/6-311++G(3*df*,3*pd*) and M06-2X/6-311++G(2*df*,2*pd*) levels of theory. The analysis of Gibbs free energy indicated that the predicted relative $\Delta G$ of MSA·MA, SFA·MA, MSA·SFA and MSA·SFA·MA clusters at the M06-2X/6-311++G(2*df*,2*pd*) level is nearly close to the values at the M06-2X/6-311++G(3*df*,3*pd*) level, with differences of less than 1.75 kcal·mol$^{-1}$. So, we chose the M06-2X/6-311++G(2*df*,2*pd*) method for further frequency calculations. Relevant details are presented in Table S7.

**Table S7** Comparison of calculated formation free energies ($\Delta G$) at the M06-2X/6-311++G(2*df*,2*pd*) and the M06-2X/6-311++G(3*df*,3*pd*) levels

| Cluster | M06-2X/6-311++G(2*df*,2*pd*) | M06-2X/6-311++G(3*df*,3*pd*) |
|---------|------------------------------|------------------------------|
|         | kcal·mol$^{-1}$ | |
| MSA·MA  | -6.19 | -6.55 |
| MSA·SFA | -9.33 | -9.54 |
| MA·SFA  | -6.01 | -6.98 |

| MSA·MA·SFA | -21.96 | -23.71 |

(c) In line 177 Page 6 to line 181 Page 7 of the revised manuscript, the reason for selecting the M06-2X/6-311++G(2$df$,2$pd$) method has been added and organized as "For all the M06-2X calculations with the 6-311++G(2$df$,2$pd$) basis set was used, as it is a good compromise between accuracy and efficiency and does not yield significant errors in the thermal contribution to the free energy compared to much larger basis sets such as 6-311++G(3$df$,3$pd$), with the differences of relative $\Delta G$ less than 1.75 kcal·mol$^{-1}$ (Table S7).".

**Comment 4.**

Page 13 lines 362-366: "Secondly, the contribution of the pathway with SFA exhibits a negative correlation with [SA] (Fig. 8 (c)), attributed to the competitive relationship between SFA and MSA. Thirdly, the contribution of the SFA-involved cluster formation pathway was positively associated with the concentration of [SFA] (Fig. 8 (d))". Rather than fixing the concentrations of other precursors and discussing the impact of changes in a single component's concentration, I think it would be more valuable to explore the specific nucleation mechanisms in regions such as India or China by incorporating observational concentrations of SFA, MSA, and MA as reported in field studies.

**Response:** Thanks for the suggestion of the reviewer. According to the reviewer's suggestion, Fig. 8(c) was redrawn to include the branching ratios of the SFA-MSA-MA (pink pie). Besides, in Lines 395-412 Page 14 of the revised manuscript, the discussion for the branching ratios of the SFA-MSA-MA has been reorganized. The main changes are as follows.

(a) To include the branching ratios of the SFA-MSA-MA, the newly revised Fig. 8(c) was redrawn and was shown in Revised Manuscript.

(b) In Lines 395-412 Page 14 of the revised manuscript, the contribution of SFA to MSA-MA system influenced by [SFA] and [MSA] has been added and reorganized as "Secondly, as depicted in Fig. 8(c) and Fig. S22, the contribution of SFA to the MSA-MA system is primarily influenced by [SFA] and [MSA], with negligible dependence on [MA]. To assess the role of SFA in MSA-MA nucleation in the atmosphere, the specific contribution of the MSA-MA cluster growth paths at varying [SFA] to NPF was calculated at 278.15 K, as illustrated in Fig. 8(c), under the ambient conditions typical of the corresponding regions. Generally, as [SFA] increases from $10^4$ to $10^8$

molecules·cm$^{-3}$, the contribution of the SFA-involved pathway increases gradually. Specifically, at low [SFA] ($10^4$ molecules·cm$^{-3}$), the contributions of SFA-involved clustering pathways are 77%

and 41% in regions with relatively low [MSA] in non-sea regions (Berresheim et al., 2002). In regions with high [SFA] ($10^6$, $10^8$ molecules·cm$^{-3}$), the contributions of the SFA-MSA-MA growth pathways are dominant in their NPF. Particularly in areas with high [MSA], such as the Pacific Rim (6.26 × $10^8$ molecules·cm$^{-3}$ (Saltzman et al., 1986)), the central Mediterranean Sea (2.11 × $10^8$

molecules·cm$^{-3}$ (Mansour et al., 2020)) and the Amundsen Sea (3.65 × $10^9$ molecules·cm$^{-3}$ (Jung et al., 2020)), nucleation is primarily driven by the SFA-MSA-MA pathway, contributing to approximately 88% of cluster formation. These results suggest that the influence of SFA is more pronounced in regions with relatively high [MSA]. It is important to note that the [SFA] values discussed in this work are estimated from limited observational data based on the reaction between

SO$_3$ and NH$_3$ in the atmosphere. Accurate determination of atmospheric [SFA] requires extensive field observations to enable more comprehensive research.".

**Comment 5.**

The boundary of the ACDC simulation is the smallest clusters that can be stable enough to grow outside of the simulated system. What's the boundary of the present ACDC simulation?

**Response.** Thanks for the suggestion of the reviewer. In ACDC simulations, boundary clusters are those allowed to flux out of the simulation box for further growth. Consequently, the smallest clusters outside the simulated system must be sufficiently stable to prevent immediate evaporation back into the system. Considering the formation Gibbs free energy (Table S7) and evaporation rates (Table S9), the clusters containing MSA and MA molecules and an SFA molecule are the most stable and are therefore allowed to grow to larger clusters, thereby contributing to the rate of NPF.

Given the above considerations, clusters (MSA)$_4$·(MA)$_3$, (MSA)$_4$·(MA)$_3$ and SFA·(MSA)$_3$·(MA)$_3$

are set as the boundary clusters for the ACDC simulation in this study. Based on the analysis above, the corresponding changes are added in Lines 193-198 Page 7 of the revised manuscript, which has been organized as "Considering the formation Gibbs free energy (Table S7) and evaporation rates (Table S9) of all clusters, the clusters containing pure MSA and MA molecules as well as the clusters containing a SFA molecule are mostly more stable and therefore are allowed to form larger clusters and contribute to particle formation rates. In this case, clusters $(MSA)_4 \cdot (MA)_3$, $(MSA)_4 \cdot (MA)_4$ and $SFA \cdot (MSA)_3 \cdot (MA)_3$ are set as the boundary clusters.".

**Comment 6.**

Page 3 line 89: "Due to the concentration of SA …, MSA-driven NPF has attracted growing attention".

Please use either "MSA" or "$CH_3SO_3H$" consistently to represent methanesulfonic acid. The same issue also appears on representation of sulfamic acid.

**Response:**

Thanks for the suggestion of the reviewer. We apologize for the misunderstanding about methanesulfonic and sulfamic acid. As the suggestion of the reviewer, the name of methanesulfonic and sulfamic acid have been corrected. Specifically, methanesulfonic and sulfamic acid has been labeled as "sulfamic acid (SFA)" and "methanesulfonic acid (MSA)", respectively, when they are first used. Besides, when they are used again, methanesulfonic and sulfamic acid has been labeled as "SFA" and "MSA", respectively.

**Comment 7.**

Page 4 line 107-108: "Atmospheric Clusters Dynamic Code (ACDC) models to evaluate the potential effect of SFA on nucleation and NPF."

Please cite the original publications of ACDC models. Additionally, cite some research to demonstrate the reliability of this method.

**Response.**

Thanks for the suggestion of the reviewer. We apologized for not referencing the original publications of ACDC models. As the suggestion of the reviewer, the original publications of ACDC models and the researches to demonstrate the reliability of this method have been cited. In Lines 107-111 Page 4, "Finally, the atmospheric implications and mechanism of SFA in the MSA-MA-dominated NPF process have been evaluated through density functional theory and the Atmospheric Clusters Dynamic Code (ACDC) models to evaluate the potential effect of SFA on nucleation and NPF." has been added as "Finally, the atmospheric implications and mechanism of SFA in the MSA-MA-dominated NPF process have been evaluated through density functional theory and the

Atmospheric Clusters Dynamic Code (ACDC) (McGrath et al., 2012; Hu et al., 2023; Zhao et al., 2020; Zhang et al., 2024; Tsona Tchinda et al., 2022; Liu et al., 2020) models to evaluate the potential effect of SFA on nucleation and NPF."

**Comment 8.**

Page 17 line 473-478: Some references include article links, while others do not. Please unify the reference format.

**Response:** Thanks for the suggestion of the reviewer. The reference format has been unified and corrected as follows:

[revised manuscript text omitted]

**Comment 9.**

The y-axis in Figure 6 contains too much information. It is recommended to adjust the layout to make the results more visually concise.

**Response.** Thanks for the suggestion of the reviewer. As the suggestion of the reviewer, the layout in Figure 6 has been adjusted to make the results more visually concise. Specifically, the sentence of "[MSA] = $10^6$, [MA] = $2.5 \times 10^8$ (molecules cm$^{-3}$)" have been removed from the Y-axis in Figure 6. The newly revised Fig. 6 is shown below.

[Figure]

Fig. 6 The $J$ (cm$^{-3}$ s$^{-1}$) (a) and $R$ (b) versus [SFA] with [MSA] = 10$^6$ molecules cm$^{-3}$, [MA] = 2.5 × 10$^8$ molecules cm$^{-3}$ and four different temperatures (green line: 298.15 K, blue line: 278.15 K, red line: 258.15 K, black line: 238.15 K).